# Web-browsing patterns reflect and shape mood and mental health

Christopher A. Kelly ●[1,2,3,4,5] ✉ & Tali Sharot[1,2,3] ✉

Humans spend on average 6.5 hours a day online. A large portion of that time is dedicated to information-seeking. How does this activity impact mental health? We assess this over four studies ($n = 1,145$). We reveal that the valence of information sought affects mental health, which in turn influences the valence of information sought, forming a self-reinforcing loop. We quantified the valence of text on webpages participants chose to browse using natural language processing and found that browsing more negative information was associated with worse mental health and mood. By manipulating the webpages browsed and measuring mood and vice versa, we show that the relationship is causal and bidirectional. Finally, we developed an intervention that altered web-browsing patterns in a manner that improved mood. Together, the findings suggest that the information sought reflects mental state and shapes it, creating a loop that may perpetuate mental health problems. The results also provide a potential method for assessing and enhancing welfare in the digital age.

Determining which factors are associated with mental health has been a key pursuit of scientists, policymakers and the general public. Research has linked mental health to various elements such as social relationships[1–3], exercise[4,5] and wealth[6,7]. In recent years, as people spend more time online, the need to investigate the relationship between online activity and mental health has become imperative[8].

One of the most common activities performed online is information-seeking[8], defined as the active pursuit of knowledge[9]. Interestingly, what people choose to know varies from one individual to the next[10–12]. These variations may provide important clues about an individual's inner cognitive and affective state[10,13,14]. For example, negative thoughts may lead to searches for information with a similar sentiment, resulting in the consumption of negatively valenced content, which could in turn exacerbate one's negative affective state. This potential mechanism is consistent with findings suggesting that people with depression tend to engage with stimuli that perpetuate their sadness[15] and is analogous to the mechanism hypothesized to underlie rumination[16,17]. Specifically, it has been suggested that continuous negative thoughts (akin to internal information-seeking) can sustain

and exacerbate low moods through a feedback loop[16,17]. In a similar vein, an individual's affective state may influence, and be influenced by, the type of information sought from external sources (as hypothesized by Sharot and Sunstein[9]). Empirically testing this hypothesis is especially important today, given the exponential increase in the availability, speed and ease of access to information, which likely amplifies the impact of information-seeking patterns on mental health.

If indeed a bidirectional relationship exists between the type of information that is consumed from self-guided web-browsing and mental health, it would have substantial theoretical and practical implications. In particular, the digital nature of online activities simplifies assessment and opens up the potential for real-time practical applications. Knowledge of the relationship between online information-seeking patterns and mental health can inform the development of tools that could complement existing interventions, such as screen time awareness tools[18,19], and digital phenotyping methods[20–26].

To date, research examining the relationship between mental health and online behaviour has predominantly focused on assessing

[1]Department of Experimental Psychology, University College London, London, UK. [2]Max Planck University College London Centre for Computational Psychiatry and Ageing Research, London, UK. [3]Department of Brain and Cognitive Sciences, Massachusetts Institute of Technology, Cambridge, MA, USA. [4]Institute for Human-Centered AI, Stanford University, Stanford, CA, USA. [5]Department of Psychology, Stanford University, Stanford, CA, USA. ✉e-mail: cakelly@stanford.edu; t.sharot@ucl.ac.uk

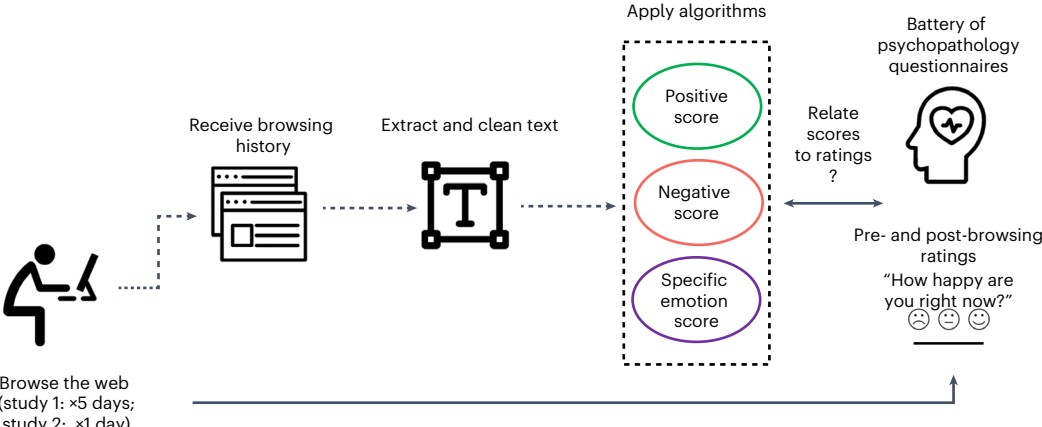

**Fig. 1 | Data collection and pre-processing pipeline.** In study 1, participants browsed the internet for 20 minutes a day for 5 days using Mozilla Firefox and then submitted their internet search history for this period. In study 2, participants browsed the internet for 30 minutes for 1 day using Mozilla Firefox and then submitted their internet search history for this period. For both studies, we extracted the paragraph text from each webpage, denoted by <p> in the webpage's html code, and cleaned it (Methods). The text was then submitted to an algorithm that calculated a negative score and a positive score for each webpage (Methods) as well as scores for anger, fear, anticipation, trust, surprise, sadness, joy and disgust (Supplementary Analysis 1). In study 1, on day 1 participants completed self-report questionnaires that assess mental health. On days 1–5, in study 1, participants also indicated their mood directly before and after each web-browsing session. In study 2, participants completed self-report questionnaires and also indicated their mood before and after the web-browsing session. Participants' scores were then related to self-reported psychopathology symptoms and mood.

screen time[27–30] and social media use. As for the latter, frequency of use has been assessed[31–34], as well as engagement styles (passive versus active[35–37]) and the type of information shared[22–26]. There have also been studies relating prevalence of specific words in Google queries (such as 'suicide', 'therapist') with indicators of the average mental health of a population[38–46].

Our approach and hypothesis are distinct from the above. We theorize that the affective properties of the information people choose to consume from self-guided browsing reflect and shape their mental health, forming a feedback loop. Our hypothesis is consistent with the rich literature showing that affect alters information-seeking and decision making[9,10,47–55]. We test this hypothesis by analysing the content an individual chooses to expose themselves to online (reflected in their web-browser history) and assessing mental health using a broad range of questionnaires. Mental health is often defined as 'emotional, psychological, and social well-being'[56], and 'mental health problems' include psychopathology disorders[57]. Following this definition, we assess mental health using self-reported questionnaires that evaluate psychopathology symptoms and experienced mood, which is an important component of 'emotional well-being'[58–61]. We note that there are variations in how different scholars define these constructs around the world[62].

Over four studies (total n = 1,145), we test the hypothesis that the affective characteristics of the information people expose themselves to online reflect and shape their subjective mental health. To quantify the affective properties of the information people expose themselves to, we asked participants to share their web-browsing history and then used a natural language processing approach to quantify the valence of the text on webpages that participants browsed. In study 1, we test whether the affective characteristics of participants' web-browsing patterns are related to their mental health. In study 2, we assess the replicability and robustness of study 1's results. Studies 3 and 4 were designed to build upon the correlational findings of studies 1 and 2, moving towards establishing causality and exploring interventions. Specifically, study 3 tests the causal direction of the relationship between mood and information-seeking, whereas study 4 investigates whether providing cues about the potential emotional impact of webpages would influence participants' web-browsing behaviour and in turn, improve their mood.

## Results

### Quantifying the affective properties of webpages

Participants in study 1 (n = 287) browsed the web for 20 minutes per day over 5 days, and those in study 2 (n = 447) for 30 minutes over 1 day. Participants then submitted their web-browsing history. We used participants' web-browsing history to access the webpages visited and extracted the text of these websites (Methods). We then scored the text on affective properties (positive and negative valence, and specific emotions) (Fig. 1). We tested three different natural language processing methods of scoring the valence of text: one large language machine-learning model (DistilBERT[63]) and two popular lexicons—the National Research Council Canada (NRC) Valence, Arousal, and Dominance (VAD) lexicon[64] and the Hu and Liu Opinion lexicon[65]. The scores of all three were highly correlated with each other, suggesting that they measure the same construct (intraclass correlation coefficient (ICC) scores were all in the range of 0.8–0.95) (Methods). Because it was found that scores calculated using the NRC lexicon reflected the valence ratings of webpages by humas a bit better than the others we opted to use it (Methods). We further showed that NRC-derived valence scores of all the text on a webpage (which we use in our analysis below) reliably reflect the valence of the sections that attract the most attention from users as well as randomly selected subsections of the text (Methods and Fig. 2).

### Quantifying mental health

Participants completed a battery of traditional psychopathology questionnaires. To quantify mental health, we adopted a dimensionality approach, which focuses on symptoms rather than diagnostic labels[10,66–70]. This nuanced approach considers the interconnected relationship between psychological conditions[66–68]. In particular, we relied on previous work that used a factor analysis across questionnaire items in a large battery of traditional psychopathology questionnaires. This analysis identified three psychopathology dimensions: 'anxious-depression', 'social-withdrawal' and 'compulsive-behaviour and intrusive thought'[66]. The factor analysis provided a weight for each item in relation to each dimension. We quantified a person's symptom severity for each of the three dimensions by calculating a weighted average across questionnaire items' ratings, which were Z-scored across participants (as done in Kelly and Sharot[10] and Seow and Gillan[68]; Methods).

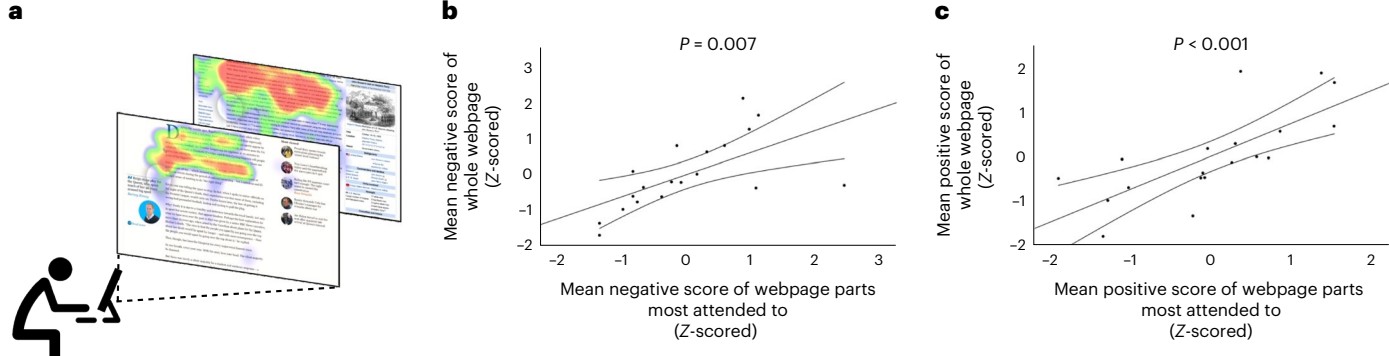

**Fig. 2 | The valence of the whole text of webpages is reflective of the valence of the text participants attend to the most. a**, Participants (*n* = 19) were asked to browse the internet for 10 minutes, while their eye movements were tracked via a web camera (https://app.gazerecorder.com). **b,c**, Our analysis included two separate mixed-effect models. **b**, The first model predicted the negative valence of the text on the entire webpage from the negative valence of the text from areas that received the most attention. **c**, The second model predicted the positive valence of the text on the entire webpage from the positive valence of the text from areas that received the most attention. Both included fixed effects and random effects and intercepts. Results showed a strong association between the valence of attended text areas (for example, highlighted in red on the heatmap) and the valence of the entire webpage text: negative valence ($\beta = 0.406 \pm 0.112$ s.e., $t(7.89) = 3.617$, $P = 0.007$) (**b**) and positive valence ($\beta = 0.304 \pm 0.051$ s.e., $t(10.47) = 5.876$, $P < 0.001$) (**c**). Thus, computing the valence of the whole text of webpages is a good indicator of the valence of text that participants consumed. Dots represent the mean negative (**b**) and positive (**c**) valence scores per individual across the webpages they visited. Outer lines represent 95% CI. Inner line represents the relationship between the abscissa and ordinate. Each dot represents a subject. All tests were two-sided.

## Browsing more negative text relates to worse mental health

We first examined whether the tendency to browse content with a specific valence was stable over time. To that end, we calculated the ICC value of the negative and positive valence of webpages visited by each participant over the 5 days. The ICC of both the negative score (ICC = 0.554, 95% confidence intervals (CI) 0.467, 0.631; *P* < 0.001) and positive score (ICC = 0.626, 95% CI 0.553, 0.690; *P* < 0.001) indicate moderate stability.

We next examined whether there is a relationship between mental health and the affective properties of the pages browsed by participants. For each participant, we calculated the three psychopathology dimension scores ('anxious-depression', 'social-withdrawal', 'compulsive-behaviour and intrusive thought'), which we submitted to a within-subjects factors mixed analysis of variance (ANOVA). In the first mixed ANOVA, the negative valence score of the webpages that participants browsed (*Z*-scored) was input as a within-subject modulating factor. Participants' age and gender were entered as between-subject modulating covariates (both *Z*-scored). We observed a significant main effect of the negative valence score of webpages participants browsed on psychopathology scores (study 1: $F(1,282) = 4.815$, $P = 0.029$, partial eta-squared = 0.017; study 2: $F(1,442) = 8.303$, $P = 0.004$, partial eta-squared = 0.018) without an interaction between the three different psychopathology scores and the negative valence score (study 1: $F(1,282) = 0.321$, $P = 0.572$, partial eta-squared = 0.001; study 2: $F(1,442) = 1.177$, $P = 0.279$, partial eta-squared = 0.003). These results indicate that individuals who browse webpages that are more negatively valenced, experience poorer mental health across the three mental health dimensions. The negative valence of webpages browsed is thus a general fingerprint of mental health, rather than associated with a specific type of condition.

To show this result in a more intuitive manner, we conducted a linear regression with psychopathology as the dependent measure (quantified as the average psychopathology score across the three dimensions) and the negative valence score of the webpages that participants browsed, age and gender as the predictor variables, all *Z*-scored. In line with the results above, we observed a significant relationship between psychopathology and the negative valence score of webpages participants browsed, with $\beta$ representing the standardized regression coefficient (study 1: $\beta = 0.091 \pm 0.042$ s.e.,

$t(286) = 2.154$, $P = 0.032$, $r = 0.127$ (Fig. 3a); study 2: $\beta = 0.099 \pm 0.034$ s.e., $t(446) = 2.930$, $P = 0.004$, $r = 0.138$, Fig. 3b), suggesting that participants who browsed more negatively valenced webpages reported worse mental health.

The second mixed ANOVA was identical to the first, except that the positive valence score was input as a within-subject modulating factor instead of the negative valence score. We did not observe a significant main effect of positive valence score of webpages on psychopathology scores in study 1 ($F(1,282) = 0.010$, $P = 0.920$, partial eta-squared = 0.000), although there was a significant effect in study 2 ($F(1,442) = 8.149$, $P = 0.005$, partial eta-squared = 0.018) with participants reporting higher psychopathology symptoms browsing less positively valenced text. We did not observe a significant interaction between the different psychopathology dimension scores (the within-subject factor) and the positive score in either study (study 1: $F(1,282) = 0.668$, $P = 0.452$, partial eta-squared = 0.002; study 2: $F(1,442) = 0.604$, $P = 0.438$, partial eta-squared = 0.001). Once again, to show this result in a more intuitive manner, we conducted a linear regression with psychopathology as the dependent measure (quantified as the average psychopathology score across the three dimensions) and the positive valence score of the webpages that participants browsed, age and gender as the predictor variables, all *Z*-scored. In line with the results above, we did not observe a significant relationship between psychopathology and the positive valence score of webpages participants browsed in study 1 ($\beta = 0.006 \pm 0.045$ s.e., $t(288) = 0.138$, $P = 0.890$, $r = 0.008$), although we did in study 2 ($\beta = -0.100 \pm 0.035$ s.e., $t(446) = -2.881$, $P = 0.004$, $r = -0.138$).

Because participants knew they would submit their browsing history, it is possible they may browse differently than they would if 'no one was watching', despite anonymity. This would induce noise that may make the relationship between web-browsing patterns and mental health more difficult to detect and thus probably even larger than reported here. Although participants were explicitly asked to browse the internet during the study session, not all of them did. We suspected as much from some of the participants short study completion times; thus, we asked participants after completing the study whether they indeed submitted data that was browsed in-session or from their archived browsing history. Thirty-nine participants in study 1 and seven in study 2 admitted they submitted archived data (because of this small

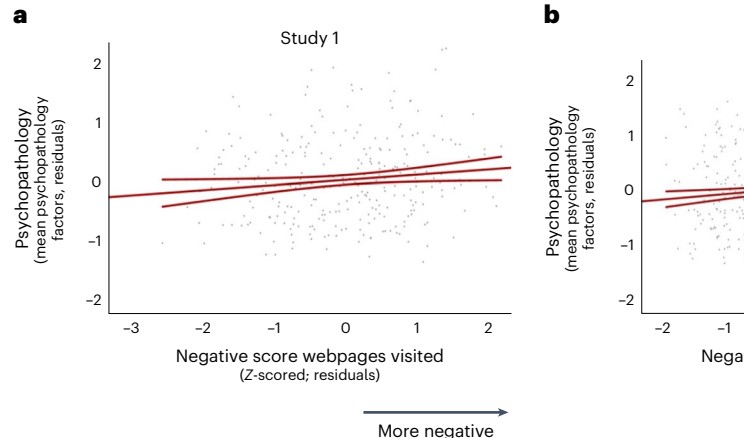

**Fig. 3 | Self-guided browsing of negative content online is associated with poorer mental health. a,b,** A linear regression analysis revealed that greater psychopathology symptoms (the average score across the three dimensions) are associated with higher negative valence scores in study 1 ($\beta$ = 0.091 ± 0.042 s.e., $t(286)$ = 2.154, $P$ = 0.032, $r$ = 0.127; $n$ = 287) (**a**) and study 2 ($\beta$ = 0.099 ± 0.034 s.e.,

$t(446)$ = 2.930, $P$ = 0.004, $r$ = 0.138; $n$ = 447) (**b**). Dots represent residual values for each participant, calculated from the model after accounting for the influence of control variables. The outer lines represent 95% CI. The inner line represents the relationship between the abscissa and ordinate controlling for the effect of age and gender. All tests were two-sided.

number the following analysis was conducted across studies 1 and 2 together). We tested whether the average valence scores of webpages browsed ($M$) from this group differed from webpages for those who browsed in-session—it was not for positive scores (browsed in study data: $M$ = 0.152, s.d. = 0.032; archived data: $M$ = 0.154, s.d. = 0.030; $t(608)$ = 0.416, $P$ = 0.678, Cohen's $d$ = 0.030) or for negative scores (browsed in study data: $M$ = 0.031, s.d. = 0.013; archived data: $M$ = 0.032, s.d. = 0.016 $t(608)$ = 0.691, $P$ = 0.498, Cohen's $d$ = 0.012). This suggests that participants were not browsing more positive or negative webpages on average because of the study set-up.

All the above results remain consistent controlling for age, gender, ethnicity, income, education and primary language spoken and after removing participants who browsed password-protected social media webpages (Supplementary Analysis 4).

**Browsing negative text relates to worse mood and vice versa**
Thus far, we observed that the extent of negative information consumed from self-guided browsing provides a general fingerprint of psychopathology symptoms, which are an indicator of poor mental health. Next, we asked whether it is also associated with mood, which has been shown to be linked to mental health[58–61], and if so whether this association is bidirectional.

To that end, we asked participants (study 1, $n$ = 164; study 2, $n$ = 400) to indicate their current mood directly before their web-browsing session and directly afterwards, on scales from 'very unhappy' to 'very happy'.

We ran linear mixed-effect models—one predicting the negative valence score and the other the positive valence score of webpages visited—from participants' pre-browsing mood ratings in study 1 (fixed and random effects) along with age and gender as fixed effects. In study 2, because we only had one observation per subject for each variable of interest (compared with five in study 1), we ran simple linear regressions predicting the negative valence score and positive valence score from pre-browsing mood ratings, controlling for age and gender (see Supplementary Analysis 4 for control variable statistics). We found that participants who reported better mood before browsing the internet, exposed themselves to less negatively valenced webpages (study 1: $\beta$ −0.082 ± 0.041 s.e., $t(380.29)$ = −1.981, $P$ = 0.048, Fig. 4a; study 2: $\beta$ = −0.097 ± 0.049 s.e., $t(396)$ = −1.974, $P$ = 0.049, $r$ = −0.099, Fig. 4a), with no significant relationship observed for positive valence score (study 1: $\beta$ = −0.021 ± 0.042 s.e., $t(104.88)$ = −0.493, $P$ = 0.623; study 2: $\beta$ = 0.088 ± 0.048 s.e., $t(396)$ = 1.830, $P$ = 0.068, $r$ = 0.092).

Next, we ran a similar analysis as above to predict post-browsing mood from the negative and positive score of webpages participants visited, while controlling for pre-browsing mood, age and gender. We found that participants expressed better mood after browsing less negatively valenced webpages in both studies (study 1: $\beta$ = −0.044 ± 0.019 s.e., $t(58.12)$ = −2.338, $P$ = 0.022, Fig. 4b; study 2: $\beta$ = −0.093 ± 0.035 s.e., $t(396)$ = −2.686, $P$ = 0.008, $r$ = −0.134, Fig. 4b). Participants also reported better mood after browsing more positive valence webpages in study 1 ($\beta$ = 0.037 ± 0.019 s.e., $t(82.62)$ = 2.013, $P$ = 0.047) but this effect was not statistically significant in study 2 ($\beta$ = 0.063 ± 0.035 s.e., $t(396)$ = 1.770, $P$ = 0.077, $r$ = 0.089). Together, these results suggest a bidirectional relationship between mood and the negative valence of webpages participants consume from self-guided browsing. Specifically, individuals that reported better mood directly before browsing the internet, browsed less negatively valenced information, and individuals who browsed less negatively valenced information reported better mood after browsing the internet.

All the above results remain consistent after controlling for age, gender, ethnicity, income, education and language and after removing participants that browsed password-protected social media webpages (Supplementary Analysis 4). Because the results are still correlational, we next ran a study to test for causation.

**Bidirectional causal association between web-browsing and mood**
In studies 1 and 2, we consistently observed a bidirectional relationship between participants' self-reported mood ratings and the negative valence scores of the webpages they browsed. This relationship was specific to negative valence scores, because no such replicable relationship was found with positive valence scores. To explore whether the former relationship is causal, we designed study 3 ($n$ = 102) to directly test the influence of browsing negatively valenced information on mood. To do so, we first manipulated the webpages participants were exposed to and tested for mood. Specifically, participants were asked to read information from two webpages randomly selected either from six negative webpages (negative valence condition, $n$ = 55) or six neutral pages (control condition, $n$ = 47). The negative pages were randomly selected from all webpages browsed in study 1 that were +2.5 s.d. from the mean negative score. The neutral webpages were randomly selected from webpages browsed in study 1 that were between −1 and +1 s.d. from the mean. Participants indicated their mood levels on a scale ranging from 'very unhappy' to 'very happy' before and after being exposed to the webpages.

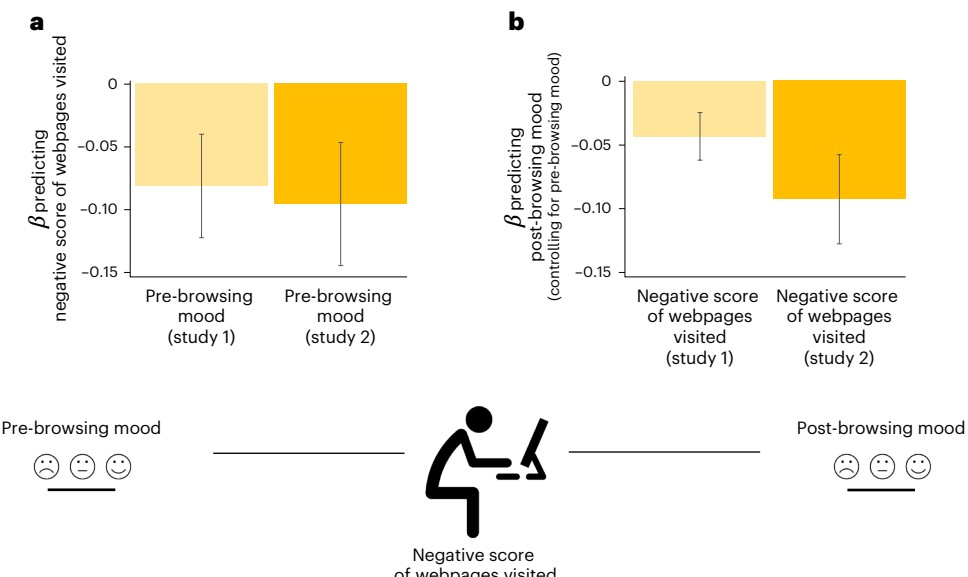

**Fig. 4 | Browsing more negatively valenced webpages is associated with worse mood before and after browsing.** In study 1 ($n = 164$), we ran a linear mixed-effect model predicting the negative valence score of webpages visited from participants' pre-browsing mood ratings (fixed and random effects) along with age and gender as fixed effects. In study 2 ($n = 400$), because we only had one observation per subject for each variable of interest (compared with five in study 1), we ran simple linear regressions predicting the negative valence score and positive valence score from pre-browsing mood ratings, controlling for age and gender. **a**, Beta coefficient predicting the negative score of webpages visited by participants from their pre-browsing mood in study 1 (light orange) and study 2 (dark orange). Participants with worse pre-browsing mood tend to browse more negatively valenced webpages controlling for age and gender in both studies (study 1: $\beta = -0.082 \pm 0.041$ s.e., $t(380.29) = -1.981$, $P = 0.048$; study 2: $\beta = -0.097 \pm 0.049$ s.e., $t(396) = -1.974$, $P = 0.049$, $r = -0.099$). **b**, Beta coefficient predicting participants post-browsing mood from the negative score of webpages they visited in study 1 (light yellow) and study 2 (dark yellow), controlling for pre-browsing mood, age and gender (study 1: $\beta = -0.044 \pm 0.019$ s.e., $t(58.12) = -2.338$, $P = 0.022$; study 2: $\beta = -0.093 \pm 0.035$ s.e., $t(396) = -2.686$, $P = 0.008$, $r = -0.134$). Participants who browsed more negatively valenced webpages reported worse post-browsing mood. Study 1, $n = 164$; study 2, $n = 400$. Error bars indicate s.e.m. All tests were two-sided.

Our rationale for manipulating and measuring mood is that mood has been shown to be easy to manipulate over short periods (minutes[71–73]), which is difficult to do for psychopathology scores[74,75]. The notion, however, is that changes in mood could eventually impact psychopathology symptoms in the long term (of which mood in itself is one[56,57]).

A linear regression analysis was conducted to predict post-browsing mood from condition (0, control condition; 1, negative valence condition), controlling for pre-browsing mood, age and gender. Condition was found to be a significant predictor of post-browsing mood ($\beta = -0.503 \pm 0.066$ s.e., $t(101) = -5.034$, $P < 0.001$, partial eta-squared = 0.151), indicating that subjects in the negative valence condition experienced lower post-browsing mood (estimated marginal mean = 6.505, s.e. = 3.059) relative to subjects in the neutral condition (estimated marginal mean = −4.787, s.e. = 2.787; Fig. 5a).

Now that the negative valence group reported worse mood than the control group, we asked whether this group of participants would go on to consume more negatively valenced webpages than the control group from self-guided browsing. To that end, participants were asked to browse the internet for 10 minutes and then submit their internet search history for this period. The negative valence of webpages participants exposed themselves to was quantified as in studies 1 and 2 (Methods). We performed an independent sample $t$-test, revealing a significant difference in the types of webpages browsed by participants based on their assigned conditions. Specifically, those participants in the negative valence condition subsequently browsed significantly more negatively valenced webpages ($M = 0.034$, s.d. = 0.020) than those in the control condition ($M = 0.026$, s.d. = 0.014, $t(96.04) = -2.259$, $P = 0.026$; Cohen's $d = -0.436$) (Fig. 5b). These results suggest a causal bidirectional relationship between participants' mood and web-browsing patterns (Fig. 5c). All results remain the consistent when controlling for age, gender, ethnicity, income, education and primary language (Supplementary Analysis 5).

### An intervention to alter patterns of web-browsing

Studies 1–3 show that browsing negatively valenced information is associated with negative symptoms of mental health. We thus pondered whether people would select to expose themselves to less negative and more positive information if they had advance knowledge of the potential affective impact of webpages. That is, would providing people with cues about the potential emotional impact of webpages alter their web-browsing patterns, resulting in less consumption of negative and more consumption of positive information?

To answer this question, we conducted study 4a. Participants were presented with three trials, each including a different Google search results page containing one search query (randomly selected from a pool of 18 queries from Google's list of frequent queries) along three real Google search results for that specific query. For each query, the three search results were selected such that one led to a webpage for which the text had a positive valence score (>2.5 s.d. from the mean positive scores of webpages browsed in studies 1 and 2), one a negative score (>2.5 s.d. from the mean negative scores of webpages browsed in studies 1 and 2) and one a neutral score (<2.5 s.d. from the mean of positive and negative scores of webpages browsed in studies 1 and 2). The presentation order of webpages was randomized to control for order effects.

On each trial, participants were to select one of the three search results offered for that specific query (we stress that all three results were regarding the same topic because they were actual result options for the same query) and would then spend 90 seconds browsing the selected webpage. Participants were notified that they would be asked a question about the content they browsed, to ensure adherence to the task.

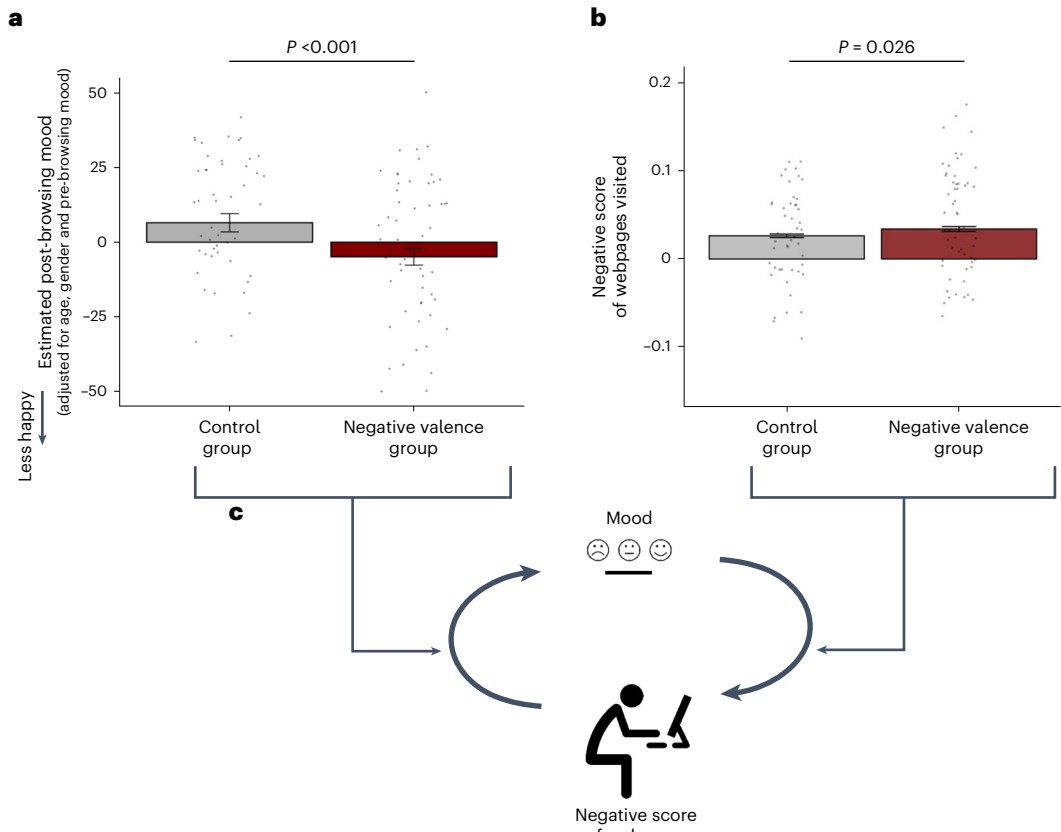

**Fig. 5 | Bidirectional relationship between mood and the valence of information consumed. a**, Participants browsed two randomly selected webpages from either six negative or six neutral (control) webpages and reported their mood before and after browsing. A linear regression analysis showed that condition (0, control; 1, negative) significantly predicted post-browsing mood, controlling for pre-browsing mood, age and gender ($\beta = -0.503 \pm 0.066$, $t(101) = -5.034$, $P < 0.001$, partial eta-squared = 0.151). The $y$ axis shows estimated post-browsing mood for both conditions, controlling for covariates. Participants in the negative condition ($n = 55$, red) experienced lower post-browsing mood (estimated mean = 6.505, s.e. = 3.059) compared with those in the control condition ($n = 47$, grey; estimated mean = −4.787,

s.e. = 2.787). Moreover, participants in the negative valence condition reported worse mood after browsing than those in the neutral condition. Individual scores are shown as dots. **b**, After browsing the webpages selected by us, participants had the opportunity to freely browse the web. Those in the negative valence condition browsed more negatively valenced webpages ($M = 0.034$, s.d. = 0.020) than those in the control condition ($M = 0.026$, s.d. = 0.014, $t(96.04) = -2.259$, $P = 0.026$; Cohen's $d = -0.436$). Individual scores are shown as dots. **c**, The results suggest a bidirectional relationship between mood and valence of webpages browsed. Specifically, worse mood leads to browsing more negatively valenced information (**b**) and browsing more negatively valenced information leads to worse mood (**a**). Error bars indicate s.e.m. All tests were two-sided.

Participants were assigned to the no-label condition or the label condition. In the label condition, participants were presented with a three-point scale next to each search result option ranging from 'feel better' to 'feel worse'. An arrow indicated where on the scale that webpage scored (either 'feel better', 'feel worse' or between the two). This indicated whether on average this website makes people feel worse/better (Fig. 6a). In the no-label condition participants were not presented with labels at all.

The question of interest was whether participants who observed the labels would choose to expose themselves to different webpages compared with participants that did not observe the labels. The results suggest they did. Specifically, participants in the label condition ($n = 55$) selected to browse webpages with negative ('feel worse') labels less than participants in the no-label condition ($n = 54$) selected to browse those same webpages ($\beta = -0.328 \pm 0.160$ s.e., $t(103) = -2.055$, $P = 0.043$, partial eta-squared = 0.041). The evidence for participants selecting webpages with positive ('feel better') labels more than participants in the no-label condition did not reach significance ($\beta = 0.348 \pm 0.177$ s.e., $t(103) = 2.966$, $P = 0.052$, partial eta-squared = 0.037). There was no difference in how often participants in the two groups selected to visit the neutral webpages ($\beta = -0.005 \pm 0.175$ s.e., $t(103) = -0.029$, $P = 0.977$, partial eta-squared = 0.000) (Fig. 6b). All results remain consistent when controlling for age, gender and ethnicity (Supplementary Analysis 5).

Our next goal was to evaluate whether the intervention influenced participants mood through their browsing choices. To explore this, we ran a follow-up study (study 4b) focused on determining whether the intervention positively affected participants' mood after browsing, while taking into account their mood before browsing. For this purpose, we recruited a new set of 200 participants to engage with the 'label' intervention. The task was as in study 4a except that participants: (1) indicated their mood at the beginning of the study (baseline) and then directly after each trial on a slider scale from 'very unhappy' to 'very happy'; (2) completed six trials rather than three; and (3) topics were randomly selected from 14 rather than 18 search queries and results (trials) from study 4a, because four websites were not accessible.

First, we found that participants selected to browse more positive webpages ($M\% = 47.75$, s.d. = 25.17) than negative ($M\% = 24.35$, s.d. = 20.02, $t(199) = 8.041$, $P < 0.001$) or neutral ($M\% = 27.90$, s.d. = 19.35, $t(199) = 6.986$, $P < 0.001$). Second, a linear mixed-effect model predicting participants mood after each trial from the valence of the webpage they selected to visit (webpages valence: −1, negative; 0, neutral; 1, positive) (fixed and random effects) controlling for their baseline (pre-intervention) mood, age and gender (fixed effects) revealed a positive relationship ($\beta = 0.090 \pm 0.023$ s.e., $t(160.74) = 3.941$, $P < 0.001$). That is, after participants exposed themselves to a webpage with more positive or less negative labels, they reported significantly better mood.

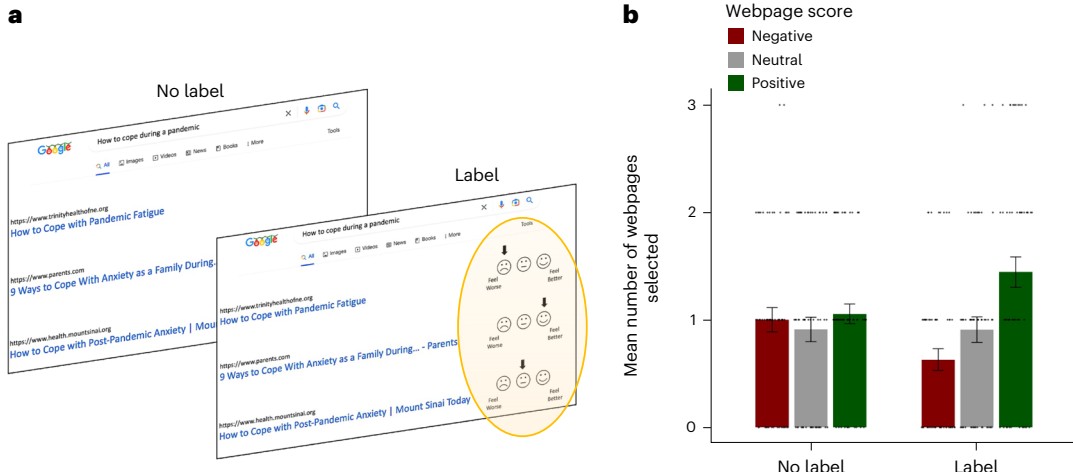

**Fig. 6 | Affective labels on search results decrease the amount of negative information visited online. a**, On each trial, participants were shown a Google search query with three actual search results (web links) below it. The topic of the Google search query was consistent across the three results. Participants chose one webpage to visit. They were assigned to either a label or no-label condition. In the label condition ($n$ = 55), a three-point scale was displayed next to each search result, ranging from 'feel better' to 'feel worse'. An arrow indicated where the webpage fell on this scale ('feel better', 'feel worse' or in between), showing how the website generally made people feel. In the no-label condition ($n$ = 54), this scale was absent. The trial illustrated here shows the first result as negative, the second as positive and the third as neutral. The yellow oval is for illustration only and was not used in the study. **b**, The bar chart presents mean number of webpages selected with a positive (green), neutral (grey) and negative (red) score

($y$ axis) by participants in the no-label and label condition ($x$ axis). Participants in the label condition clicked fewer negative webpages ($\beta$ = −0.328 ± 0.160 s.e., $t$(103) = −2.055, $P$ = 0.043, partial eta-squared = 0.041) than those in the no-label condition. The evidence for participants selecting webpages with positive ('feel better') labels more than participants in the no-label condition did not reach significance ($\beta$ = 0.348 ± 0.177 s.e., $t$(103) = 2.966, $P$ = 0.052, partial eta-squared = 0.037). Also, there was no significant difference in selecting neutral webpages ($\beta$ = −0.005 ± 0.175 s.e., $t$(103) = −0.029, $P$ = 0.977, partial eta-squared = 0.000). This suggests that affective cues about webpages alter browsing behaviour, reducing exposure to negative content. Individual scores are shown as dots. All tests controlled for age and gender and were two-sided. Error bars indicate s.e.m.

This implies that providing cues regarding the emotional properties of webpages can help users make choices that improve mood. All results remain consistent after controlling also for ethnicity, income, education and primary language, except that participants in the label condition also significantly browsed more positive labels than in the 'no-label' condition ($\beta$ = 0.364 ± 0.182 s.e., $t$(102) = 3.179, $P$ = 0.048; Supplementary Analysis 5).

## Discussion

Our findings reveal that web-browsing reflects and shapes mental health. Individuals who consumed more negative information from self-guided web-browsing reported worse mental health and mood. In addition, participants who reported worse mood before browsing tended to access more negative content online. This exposure to negative content was, in turn, associated with worse mood after browsing (controlling for mood before browsing).

We established the causality of this relationship by exposing participants to either negative or neutral webpages. We found that exposure to negative webpages resulted in worse mood, and this worsening of mood then led people to browse more negatively valenced information. Together, these findings reveal a feedback loop; low mood leads to the consumption of more negative information, which in turn, leads to worse mood and so on.

The results contribute to the ongoing debate regarding the relationship between mental health and online behaviour (for a review see refs. 76,77). Most research addressing this relationship has focused on the quantity of use, such as screen time or frequency of social media use, which has led to mixed conclusions[27–34]. Here instead, we focus on the type of content browsed and find that its affective properties are causally and bidirectionally related to mental health and mood. A key advancement of this approach is the analysis of the text on webpages visited, rather than focusing on screen time or analysing queries entered into a search engine (for example, as done in refs. 38,39). By analysing the text on webpages, we obtain a rich dataset that goes

beyond a limited set of keywords (for example, therapist or Prozac) associated exclusively with individuals who are explicitly seeking help.

The findings are consistent with studies showing a relationship between the affective properties of the content people share and their mental health (such as posting on social media[24–26,78]), which may indicate an intriguing overlap between the mechanism governing information-seeking and information-sharing[79]. Indeed, the size of the effects reported here are comparable with those reported in those studies[25], as well as in studies using other methods[33,80–83]. One advantage of examining the information consumed, is that it does not necessitate that people share information online, an activity that is clearly prevalent, but less so than information-seeking.

Once we had established a causal relationship between the consumption of negative information online and negative mood, we examined whether individuals would choose to access less negative information if they were made aware of the potential emotional impact of the content on webpages. Indeed, our results showed that providing individuals with cues about the valence of webpages led them to browse less negative content, which improved their mood. These results suggest that a simple intervention—namely providing labels that inform users in advance of the affective properties of webpages they consider browsing—is effective in reducing exposure to negative information and improving mood.

In many cases, however, it would be suboptimal to solely base information-consumption decisions on the affective properties of information. For example, if someone searches for information on the relationship between smoking and cancer, negatively valenced content may be lifesaving. Thus, we do not recommend an intervention in which only 'affective labels' are provided. Rather, we propose focusing on three characteristics of information that are key for people's information-seeking decisions[9]: (1) the capacity of information to guide actions[9,10,53,55,84]; (2) the potential of information to expand epistemic knowledge[9,10]; and (3) the valence of the information[9,10,55]. For example, we recently developed a tool[85] in the form of a plug-in that presents

users with three scores alongside each webpage link on a search engine result page. The scores are calculated using a machine-learning algorithm that estimates the actionability of the text on that webpage, the valence of the text and the likelihood of it to expand knowledge. By utilizing these scores, users can tailor their web-browsing experience to match their current goals (for example, improving mood, finding actionable information).

## Limitations

The current results point to a consistent relationship between mental health symptoms and the consumption of negatively valenced text, rather than positively valenced text. This observation is consistent with past studies that find that individuals with poor mental health are more likely to attend to negative information[86,87] and use it to alter their beliefs[88]. Although it is possible that the non-significant relationship between mental health and a tendency to consume positive information from self-guided browsing reflects a ground truth, it is also possible that a small effect exists that we failed to pick up. For example, if participants intentionally adjusted their behaviour (that is, not including all the webpages they actually visited or not sincerely completed the mental health questionnaires), this would induce noise that would make smaller effects difficult to detect. Steps can be taken in the future to enhance the methodology used here to increase the likelihood of detecting such effects. Improvements can include adding analysis of images and videos; collecting timestamps of participants' web-browsing to measure the exact amount of time users spend on each piece of content; including password-protected websites such as social media platforms; including browsers beyond Mozilla Firefox, and extending the duration of data collection to weeks or months while using ecological momentary assessment. The latter allows us to characterize the relationship between mental health and web-browsing patterns at a more granular level. Moreover, although we used the NRC lexicon to measure valence because it showed the strongest relationship to human raters compared with other lexicons and large language models, NRC does not incorporate context.

It is also interesting to consider the role of recommendation algorithms here. Many algorithms are trained on participants' past behaviour, thus might perpetuate a participant's affective state by promoting specific types of information (such as negatively valenced content), potentially exacerbating the feedback loop identified. Moreover, algorithms may also be a source of noise, in the sense that they alter peoples' natural search intentions. If that is the case, the relationship between mental health and self-guided browsing is likely even greater than we report here.

## Conclusion

We developed a methodology—analysing the content browsed—to explore the causal bidirectional relationship between mental health and web-browsing patterns. Our approach combines psychological theory with computer science to both advance theoretical understanding and inform the development of practical tools. The results show that the affective properties of the information people choose to consume from self-guided browsing reflects and shapes their mental health, forming a feedback loop. These findings led to the development of a tool that helps users browse the web in an informed manner and that can enhance mood and may improve mental health.

## Methods

Ethical approval was provided by the Research Ethics Committee at University College London and all participants gave their informed consent to participate. Each subject participated in only one of the studies reported.

### Study 1

**Participants.** Three hundred and twelve participants were recruited through the Prolific platform. Participants were recruited from the UK and US and were 18 years or older. Sample sizes were determined based on a pilot study to achieve a power of 0.95 ($\alpha = 0.05$), using G*Power[89]. Data from 25 participants whose browsing did not result in at least 1 KB of text from at least three webpages each day were not analysed further. Thus, data for 287 participants were analysed (age, 33.17 years, s.d. 11.71; 50.5% females, 48.1% males, 1.4% other). Of those, 171 participants also completed state mood ratings. Data of five participants who indicated that they submitted archived browsing history were not included in mood analysis, because their current mood ratings obviously could not be temporally associated with their submitted browsing data, leaving $n = 164$ for mood analysis (age, 33.23 years, s.d. 11.62; 52.4% females, 47.6% males, 0% other). All participants received £7.50 for their participation on day 1 and £3.25 on each of days 2–5.

**Procedure.** *Data collection.* Participants were asked to browse the internet for 20 minutes a day for 5 days using Mozilla Firefox and then to submit their internet search history for this period (see Supplementary Materials for study instructions). We used Mozilla Firefox because it was the only browser, to our knowledge, that allowed users to extract the exact URL they visited with relative ease from their browsing history. We extracted the paragraph text from each webpage, denoted by <p> in the webpage's html code, using the 'rvest' package in RStudio. We then cleaned the text by removing extraneous information such as punctuation, symbols (for example, @, #), emojis, links (URLs) and all other non-alphanumeric characters (similar to Kelley and Gillan[22]). Participants were asked to browse the internet during non-work hours so that their web-browsing behaviour would not reflect mandatory work-related tasks. All consecutive duplicate webpages were removed from the analysis. Being mindful that content can change quickly, we made sure to extract the vast majority of the text from webpages within 24 hours (and at the very most 36 hours) of the time the participant visited the page. However, it is still possible that some content may have changed during this time. Therefore, we ran a validation to check for the stability of webpages' valence scores (see Supplementary Analysis 3 for details). This validation confirmed that the valence of webpages remain highly stable across consecutive days (negative score: $r(998) = 0.991$, 95% CI 0.990, 0.992, $P < 0.001$; positive score $r(998) = 0.991$, 95% CI 0.989, 0.992, $P < 0.001$).

*Selecting the method to measure text valence.* There are many validated methods to score text on sentiment (valence). These include machine-learning methods[90–92] and 'bag of words' (lexicon) approaches that are developed by asking large groups of people to rate words on specific dimensions[64,65]. We first tested whether these different methods provide consistent scores for participants. We selected two popular lexicons—the NRC VAD lexicon[64] and the Hu and Liu Opinion lexicon[65]—and a state-of-the-art large language machine-learning model, the distilbert-base-uncased-finetuned-sst-2-english (Distil-BERT[63]), which is fine-tuned for sentiment analysis tasks. For each webpage, the DistilBERT model provides probabilities representing the likelihood of the text expressing positive or negative sentiment. These probabilities were the positive and negative sentiment scores used for the large language model. For the NRC VAD lexicon[64], the valence of each word is categorized on a scale from 0 (most negative) to 1 (most positive). In line with Kiritchenko and colleagues[93], we computed the percentage of words with a positive valence score ≥0.75 (2,668 terms; for example, 'delicious' and 'admire') and percentage of words with a negative valence score ≤0.25 (3,081 terms; for example, 'despise' and 'danger'), of all words contained in the extracted text of each webpage visited for each of the 5 days. For the Hu and Liu method[65], the lexicon contains separate positive and negative word lists, with no weighting applied as in the NRC method. We calculated the percentage of positive and negative words from all words contained in the extracted text of each webpage visited over the same 5-day period.

We used each method separately to score all webpages visited by the first 100 participants from study 1 and averaged the webpage scores

for each participant. We used an ICC analysis to examine how consistent the scores were across different scoring methods, separately for positive and negative scores. All scores were $Z$-scored before analysis to ensure they were on the same scale. We observed good reliability between all three methods: (1) the NRC VAD lexicon and the Hu and Liu Opinion lexicon (positive score: ICC = 0.835, $P < 0.001$; negative score: ICC = 0.948, $P < 0.001$); (2) the NRC VAD lexicon and the DistilBERT algorithm (positive score: ICC = 0.812, $P < 0.001$; negative score: ICC = 0.869, $P < 0.001$); and (3) the DistilBERT algorithm and the Hu and Liu Opinion lexicon (positive score: ICC = 0.866, $P < 0.001$; negative score: ICC = 0.885, $P < 0.001$). This suggests that these different methods measure the same construct.

We further checked that the above scores were reflective of human assessment. To that end, we asked a fresh set of participants ($n = 100$) to rate the positive (0 (not at all) to 6 (very positive)) and negative (0 (not at all) to 6 (negative)) valence of 10 randomly assigned webpages from a corpus of 48 webpages. We then computed the positive and negative valence scores for each webpage using the methods above and their respective human rating for that webpage and submitted the positive and negative pairs into an ICC to calculate their reliability. All scores were $Z$-scored to standardize them for comparison. The values were found to be significantly related: the human ratings were significantly related with the NRC valence scores (negative score: ICC = 0.707, 95% CI 0.668, 0.742, $P < 0.001$; positive score: ICC = 0.499, 95% CI 0.432, 0.558, $P < 0.001$), with the Hu and Liu valence scores (negative score: ICC = 0.680, 95% CI 0.668, 0.742, $P < 0.001$; positive score: ICC = 0.510, 95% CI 0.432, 0.558, $P < 0.001$) and with the DistilBERT algorithm scores (negative score: ICC = 0.472, 95% CI 0.402, 0.534, $P < 0.001$; positive score: ICC = 0.384, 95% CI 0.302, 0.465, $P < 0.001$). Because the NRC lexicon was found to reflect human subjective assessments of webpages a bit better on average compared with both the Hu and Liu method and the machine-learning method, and it also has a specific emotion lexicon while requiring fewer computational resources than the machine-learning methods, we chose it for our analysis.

Given that the method we used scores entire webpages rather than the text participants actually consume, it was important to test whether the former was a good indicator of the latter. To that end we adopted two approaches. First, we examined whether there is good reliability between the valence of text on a whole webpage and the valence of text on a random part of it. To test this, we randomly extracted segments of text from webpages ($n = 100$) with a minimum word count of 200 words[94]. We then calculated the positive and negative scores for the random samples text and that of its corresponding whole text and submitted those into an ICC analysis to calculate reliability (separately for the positive and negative scores). We observed good reliability between the NRC valence scores of randomly sampled segments and the scores of their respective webpage's whole texts (negative score: ICC = 0.945, 95% CI 0.918, 0.963, $P < 0.001$; positive score: ICC = 0.947, 95% CI 0.922, 0.965, $P < 0.001$). This result suggests that by analysing the whole text of a webpage, we can reliably compute the sentiment of a random section of a webpage.

Second, we examined directly whether there is good reliability between the valence scores of the text of a whole webpage and the valence of the text that participants attended to the most. To test this, a new group of participants were asked to browse the internet for 10 minutes, while their eye movements were tracked via a web camera (https://app.gazerecorder.com) (Fig. 2a). This test involved 19 participants who collectively visited 59 different websites. Participants were included in the study if they visited at least one webpage that contained paragraph text. We calculated the NRC valence scores for both the text areas that captured most of the participants' attention (highlighted in red on the heatmap generated by our algorithm in Fig. 2a) and the entirety of the text on each webpage. Our analysis included two separate mixed-effect models (using the 'lme4' package in RStudio). The first model predicted the positive valence of the text on the entire

webpage from the positive valence of the text from areas that received the most attention. The second model predicted the negative valence of the text on the entire webpage from the negative valence of the text from areas that received the most attention. Both included fixed effects and random effects and intercepts. The results showed that both the positive valence ($\beta = 0.304 \pm 0.051$ s.e., $t(10.47) = 5.876$, $P < 0.001$) and negative valence ($\beta = 0.406 \pm 0.112$ s.e., $t(7.89) = 3.617$, $P = 0.007$) of the attended to text areas were strongly associated with the valence of the entire webpage text (Fig. 2b,c). Therefore, the overall valence scores of a webpage's text can reliably reflect the valence of the sections that attract the most attention from users.

Next, we conducted two studies to test whether key parameters are different based on device used (Supplementary Analysis 2). First, we asked one group of participants ($n = 25$) to view webpages on their smartphones and another group ($n = 25$) to view the same webpages on their desktop or laptop computers. All were to provide two sentiment ratings (positive and negative) of each page on a six-point Likert scale from 'not at all' to 'very much' and compared scores across devices. Second, we asked a new group of participants ($n = 28$) to browse the internet for 15 minutes on their mobile phone on one day and on their desktop or laptop on another day. We then calculated the valence scores of the webpages they selected to browse on each day and compared across devices. The findings from these two studies demonstrate that the valence and affective impact of webpages do not differ based on the device used for browsing, whether it be a mobile phone or a desktop or laptop (Supplementary Analysis 2).

*Assessment of mental health and mood.* On day 1, before the web-browsing task, participants completed self-report questionnaires that assess psychopathology symptoms (the list is adopted from Gillan and colleagues[66]) These were: Obsessive-Compulsive Inventory−Revised[95], Self-Rating Depression Scale[96], State−Trait Anxiety Inventory[97], Alcohol Use Disorder Identification Test[98], Apathy Evaluation Scale[99], Eating Attitudes Test[100], Barratt Impulsivity Scale[101], Short Scales for Measuring Schizotypy[102] and Liebowitz Social Anxiety Scale[103]. On days 1–5, participants indicated their current mood directly before their web-browsing session and directly afterwards, on a scale from 'very unhappy' to 'very happy'. Happiness is considered a key component of overall well-being[104] and mental health[62]. Indeed, the American Psychological Association (APA) definition of mental health states that emotional well-being, which includes happiness, is an integral part of mental health[62]. The use of a continuous scale for assessing levels of happiness is a well-established method in psychological research[105–107]. This allowed us to test whether participants' pre-browsing mood and post-browsing mood was related to the valence of information they browsed. The task was coded using the Qualtrics online platform (https://www.qualtrics.com).

**Analysis.** *Assessing the stability of the valence of web-browsing across time.* To assess the within-subject stability of the valence of webpages visited across the 5 days, we calculated an ICC. Specifically, we submitted separately the negative and positive valence score and the scores for the specific emotions of webpages visited by each participant for each of the 5 days into ICC analysis.

*Relating the valence of webpages to mental health.* Each participant was scored on the three psychopathology dimensions identified by Gillan and colleagues[66] and replicated by Rouault and colleagues[67] ('anxious-depression', 'social-withdrawal' and 'compulsive-behaviour and intrusive thought'). To generate these scores, we followed Kelly and Sharot[10] and Seow and Gillan[68]—we first $Z$-scored the ratings for each questionnaire item separately across participants. Next, we multiplied each $Z$-scored item by its factor weight as identified earlier[66]. Then for each subject the three psychopathology dimension scores were calculated by summing all the weighted items assigned to each dimension.

For each participant, we calculated the positive valence and negative valence scores separately across all webpages visited on each day and then averaged the daily scores across the 5 days to create a positive valence score and negative valence score, respectively. We also quantified separately the percentage of anger, fear, anticipation, trust, surprise, sadness, joy and disgust associated words ≥0.75, as defined by the NRC Emotion Lexicon[108], of all words on each webpage visited by participants for each day and then across days (emotion scores; see Supplementary Analysis 1 for details).

We then related the psychopathology dimensions scores to each affective score separately by submitting the three psychopathology dimension scores into a mixed ANOVA with psychopathology dimension as a within-subject factor and the valence score as within-subject modulating covariates as well as participants' age and gender as between-subjects modulating covariates (similar to ref. 10). This analysis was followed up with a simplified analysis in which the average of the three psychopathology dimension scores for each individual was entered as a dependent measure in a linear regression with valence entered as an independent measure as well as age and gender. The data met assumptions for stated statistical tests.

For statistical analysis we used a combination of RStudio and SPSS. For text pre-processing and quantification, we used a combination of RStudio and Python (same for all studies).

*Relating the valence of webpages to mood.* To investigate the relationship between web-browsing patterns and mood, we asked participants to indicate their current mood directly before their web-browsing session and directly afterwards, on a slider scale from 'very unhappy' to 'very happy'. Utilizing a continuous scale is a common method for evaluating happiness levels[105–107]. We first assessed whether participants pre-browsing mood was related to the valence of information they browsed. To that end, we ran two separate mixed-effect models each including participants pre-browsing mood ratings (which we coded by converting the scale to −50 to +50) as fixed and random effects along with age and gender as fixed effect predicting the negative valence score and positive valence score of webpages visited, separately. Next, we were interested in whether the valence of the webpages that participants browsed had an impact on their mood directly after browsing the internet. To test this, we once again ran two mixed-effect models, each predicting post-browsing mood ratings (which we coded by converting the scale to −50 to +50) from either the negative and positive valence score of webpages visited (input as a fixed and random effect), controlling for pre-browsing mood (fixed and random effect) as well as age and gender (fixed effect). The data met assumptions for stated statistical tests.

**Study 2: replication of study 1**
**Participants.** Five hundred participants were recruited through the Prolific platform. Sample sizes were determined based on a power analysis relating negative score of web-browsing on day 1 from study 1 with mean psychopathology scores (G*Power[89]: $\alpha = 0.05, 1 - \beta = 0.95$). The majority of participants were recruited from the UK and US, with a subset ($n = 168$) recruited from any country without restriction. Participants were 18 years or older. There was no other inclusion or exclusion criteria. Data for 53 participants from whom we could not obtain at least 1 KB of text from a minimum of three webpages a day was not analysed. Thus, data for 447 participants were analysed (age 33.85 years, s.d. 12.58; 56.4% females, 41.8% males, 1.8% other). For the mood analysis, we included only those participants who submitted data that was browsed during the study session ($n = 400$, age 33.23 years, s.d. 11.62; 52.4% females, 47.6% males, 0% other), because otherwise their reported mood ratings would not be temporally reflective of their submitted browsing data. Participants received £7.50 for their participation.

**Procedure.** Study 2 replicated the methodology of study 1 with two modifications. First, we required participants to engage in a 1-day, 30-minute internet browsing session. This decision was made after a post hoc analysis of study 1, aimed at achieving a balance between statistical rigour and resource efficiency, including cost-effectiveness. Our analysis suggested that involving approximately 500 participants for this single-day study could lead to substantial cost reductions. In addition, we slightly extended the data collection time frame to gather adequate data per participant. Second, in contrast to study 1, psychopathology questionnaires were administered post web-browsing to show that the sequence of tasks in study 1 did not influence its outcomes. Mood ratings were still recorded before and after the session.

**Analysis.** *Relating the valence score of webpages to psychopathology.* This analysis was conducted as described in study 1.

*Relating the valence of webpages to mood.* We first tested whether participants pre-browsing mood was related to the valence of information they browsed. Because we had only one observation per participant for each variable of interest (compared with five observations in study 1), we ran two simple linear regressions predicting the negative valence score and positive valence score, separately, from pre-browsing mood ratings, controlling for age and gender. Next, we were interested in whether the valence of the webpages that participants browsed had an impact on their mood directly after browsing the internet. To test this, we ran two simple linear regressions, both predicting participants post-browsing mood ratings from either the negative or positive valence score of webpages visited. Both models controlled for participants pre-browsing mood ratings, age and gender. The data met assumptions for stated statistical tests.

**Study 3**
**Participants.** One hundred and thirty-nine participants completed the study on Qualtrics (www.qualtrics.com) and were recruited via Prolific's online recruitment platform (www.prolific.co). Sample sizes were determined based on a pilot study to achieve a power of 0.95 ($\alpha = 0.05$), using G*Power[89]. Participants received £7.50 per hour for their participation. Participants were recruited from the UK and US and were 18 years or older. There were no other inclusion or exclusion criteria. Thirty-seven participants were excluded for not providing at least three webpages from which we could extract at least 1 KB of data, leaving 102 participants (negative valence condition: $n = 55$, age 33.96 years, s.d. 9.68; 45.5% females, 49.1% males, 5.5% other; control condition: $n = 47$, age 34.72 years, s.d. 12.14; 46.8% females, 51.1% males, 2.1% other). The two conditions were run within 35 minutes of each other. We did not actively randomize participants across conditions, rather the recruitment advertisement for the conditions was identical for both conditions. This means the participants were not aware which condition they were signing up for, nor were they aware that there were two conditions, ensuring no differences in demographics across groups (Supplementary Table 1). Stimuli presentation was random.

**Procedure.** *Data collection.* Here, we assess the directionality of the relationship between mood and web-browsing patterns. Participants were asked to browse two webpages randomly selected from a pool of either six very negative or six neutral webpages (all selected from pages participants browsed in study 1 or 2). The negative webpages were identified based on a negative score of >2.5 s.d. from the mean of webpages browsed in studies 1 and 2, whereas the neutral webpages had a negative score ranging between −1 to +1 s.d. from this mean. The selection criteria for the valence of the webpages were consistent with the methodology outlined in study 1. Participants' happiness levels were measured on a scale, from 'very unhappy' to 'very happy', both before and after the webpage manipulation. Utilizing a continuous scale is a widely used method for evaluating happiness levels[105–107].

Next, participants were asked to browse the internet for 10 minutes using Mozilla Firefox and then submit their internet search history for this period. We then extracted the paragraph text from each webpage, denoted by <p> in the webpage's html code, using the 'rvest' package in RStudio. All consecutive duplicate webpages were removed from analysis.

**Analysis.** To assess whether the mood manipulation was successful, we conducted a linear regression predicting participants post-browsing mood ratings from the condition variable (0, neutral condition; 1, negative condition), controlling for pre-browsing mood, age and gender. Next, for each participant, we computed the negative valence score of the webpages browsed. Finally, we conducted an independent sample $t$-test to investigate whether there is a difference in the negative valence score of the webpages browsed between the conditions. The data met assumptions for stated statistical tests.

### Study 4a
**Participants.** One hundred and nine participants (label condition: $n = 55$, age 36.94 years, s.d. 13.68; 67.7% females, 33.3% males, 0% other; no-label condition: $n = 54$, age 36.09 years, s.d. 9.97; 49.1% females, 47.3% males, 3.6% other) completed the study on Qualtrics (www.qualtrics.com) and were recruited via Prolific's online recruitment platform (www.prolific.co). Sample sizes were determined based on a pilot study to achieve a power of 0.95 ($\alpha = 0.05$), using G*Power[89]. Participants were recruited from the UK and US and were 18 years or older. There were no other inclusion or exclusion criteria. Participants received £7.50 per hour for their participation. Twenty-three participants were recruited 1 day before the main data collection to assure the negative mood manipulation was working, otherwise the two conditions were run within 2 hours of each other.

We did not actively randomize participants across conditions, rather the recruitment advertisement for the conditions was identical for both conditions. This means the participants were not aware which condition they were signing up for, nor were they aware there were two conditions, ensuring no differences in demographics across groups (Supplementary Table 1). Stimuli presentation was random.

**Procedure.** *Data collection*. Participants were presented with three trials, each including a different Google search results page containing one search query (randomly selected from a pool of 18 queries from Google's list of frequent queries) along three real Google search results for that specific query. For each query, the three search results were selected such that one led to a webpage for which the text had a positive valence score (that is >2.5 s.d. from the mean positive scores of webpages browsed in studies 1 and 2), one a negative score (>2.5 s.d. from the mean negative scores of webpages browsed in studies 1 and 2), and one a neutral score (<2.5 s.d. from the mean of positive and negative scores of webpages browsed in studies 1 and 2). The presentation order of webpages was randomized to control for order effects.

On each trial participants were to select one of the three search results offered for that specific query (we stress that all three results were regarding the same topic because they were actual result options for the same query) and then spent 90 seconds browsing the selected webpage. Participants were notified that they would be asked a question about the content they browsed, to ensure adherence to the task.

Participants were assigned to the no-label condition or the label condition. In the label condition participants were presented with a three-point scale next to each search result option that went from 'feel better' to 'feel worse'. An arrow indicated where on the scale that webpage scored (either 'feel better', 'feel worse' or between the two). The labels indicate whether on average this website makes people feel worse or better (Fig. 6a). In the no-label condition participants were not presented with any emojis or labels next to the search results.

**Analysis.** To assess whether the intervention was successful, we conducted separate linear regressions each predicting either the mean number of positive, neutral and negative labels selected by participant from the condition variable (0, no-label condition; 1, label condition), controlling for age and gender. The data met assumptions for stated statistical tests.

### Study 4b
**Participants.** Two hundred participants (age 40.8 years, s.d. 12.9; 58.0% females, 40.5% males, 1.5% other) completed the study on Qualtrics (www.qualtrics.com) and were recruited via Prolific's online recruitment platform (www.prolific.co). Sample sizes were determined based on a pilot study to achieve a power of 0.95 ($\alpha = 0.05$), using G*Power[89]. Participants were recruited from the UK and US and were 18 years or older. There were no other inclusion or exclusion criteria. Participants received £9.00 per hour for their participation. Stimuli presentation was random.

**Procedure.** The task was exactly as in study 4b label condition, except that participants: (1) indicated their mood at the beginning of the study (baseline) and then directly after each trial on a slider scale from 'very unhappy' to 'very happy'; (2) completed six trials rather than three; and (3) topics were randomly selected from 14 rather than 18 search queries and results (trials) from study 4a (because four trials were not accessible).

**Analysis.** First, we conducted a paired samples $t$-test comparing the number of times participants selected the positive label compared with the neutral and negative label. Next, a linear mixed-effect model was implemented to analyse the impact of webpage choice on mood. Webpage choice, categorized as negative ($-1$), neutral (0) or positive (1), served as the independent variable (fixed and random effects). The model also controlled for baseline mood, age and gender as fixed effects to account for any confounding influences. This approach allowed us to isolate the direct effect of the independent variable—webpage choice—on the mood of the participants. The data met assumptions for stated statistical tests.

### Reporting summary
Further information on research design is available in the Nature Portfolio Reporting Summary linked to this article.

### Data availability
Anonymized data is available via GitHub at https://github.com/affective-brain-lab/WebbrowsingNHB.

### Code availability
Code is available via GitHub at https://github.com/affective-brain-lab/WebbrowsingNHB.

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

## Acknowledgements

We thank L. Globig, B. Blain, M. Glickman, T. Nahari, V. Vellani, S. Zheng, I. Pinhorn, H. Haj-Ali, J. Serfaty, N. Markovitch, C. Moutsiana and D. Rebbin for comments on previous versions of the manuscript. The work was funded by a Wellcome Trust Senior Research Fellowship (grant no. 214268/Z/18/Z) awarded to T.S. The funders had no role in study design, data collection and analysis, decision to publish or preparation of the manuscript.

## Author contributions

C.A.K. and T.S. designed the study. C.A.K. collected and analysed the data with guidance from T.S. C.A.K. and T.S. drafted the manuscript. Both authors approved the final version of the manuscript for submission.

## Competing interests

The authors declare no competing interests.

## Additional information

**Correspondence and requests for materials** should be addressed to Christopher A. Kelly or Tali Sharot.

# Reporting Summary

## Statistics

For all statistical analyses, confirm that the following items are present in the figure legend, table legend, main text, or Methods section.

| n/a | Confirmed | |
|---|---|---|
| ☐ | ☒ | The exact sample size (*n*) for each experimental group/condition, given as a discrete number and unit of measurement |
| ☐ | ☒ | A statement on whether measurements were taken from distinct samples or whether the same sample was measured repeatedly |
| ☐ | ☒ | The statistical test(s) used AND whether they are one- or two-sided *Only common tests should be described solely by name; describe more complex techniques in the Methods section.* |
| ☐ | ☒ | A description of all covariates tested |
| ☐ | ☒ | A description of any assumptions or corrections, such as tests of normality and adjustment for multiple comparisons |
| ☐ | ☒ | A full description of the statistical parameters including central tendency (e.g. means) or other basic estimates (e.g. regression coefficient) AND variation (e.g. standard deviation) or associated estimates of uncertainty (e.g. confidence intervals) |
| ☐ | ☒ | For null hypothesis testing, the test statistic (e.g. $F$, $t$, $r$) with confidence intervals, effect sizes, degrees of freedom and $P$ value noted *Give P values as exact values whenever suitable.* |
| ☒ | ☐ | For Bayesian analysis, information on the choice of priors and Markov chain Monte Carlo settings |
| ☒ | ☐ | For hierarchical and complex designs, identification of the appropriate level for tests and full reporting of outcomes |
| ☐ | ☒ | Estimates of effect sizes (e.g. Cohen's *d*, Pearson's *r*), indicating how they were calculated |

*Our web collection on statistics for biologists contains articles on many of the points above.*

## Software and code

Policy information about availability of computer code

| Data collection | The experiments were designed using Qualtrics' online survey platform. Participants were recruited using Prolific. Participants were asked to browse the web using Firefox. |
|---|---|
| Data analysis | IBM SPSS 27, R studio Version 2022.12.0+353, Google Colab (i.e., Python). |

For manuscripts utilizing custom algorithms or software that are central to the research but not yet described in published literature, software must be made available to editors and reviewers. We strongly encourage code deposition in a community repository (e.g. GitHub). See the Nature Portfolio guidelines for submitting code & software for further information.

## Data

Policy information about availability of data

All manuscripts must include a data availability statement. This statement should provide the following information, where applicable:
- Accession codes, unique identifiers, or web links for publicly available datasets
- A description of any restrictions on data availability
- For clinical datasets or third party data, please ensure that the statement adheres to our policy

Data and Code availability: Anonymized data and code are available at a dedicated repository [https://github.com/affective-brain-lab/WebbrowsingNHB].

# Research involving human participants, their data, or biological material

Policy information about studies with [human participants or human data](). See also policy information about [sex, gender (identity/presentation), and sexual orientation]() and [race, ethnicity and racism]().

| | |
|---|---|
| Reporting on sex and gender | We only inquired about Gender. Necessary information has been reported in the manuscript and below. |
| Reporting on race, ethnicity, or other socially relevant groupings | In addition to gender we report on age, ethnicity, income, education and language. |
| Population characteristics | See below. |
| Recruitment | Participants were recruited using Prolific's online recruitment platforms. The purpose of this study was not mentioned in the advertisement of the study. |
| Ethics oversight | Studies were approved by the ethics committee at UCL and all subjects gave informed consent. |

Note that full information on the approval of the study protocol must also be provided in the manuscript.

# Field-specific reporting

Please select the one below that is the best fit for your research. If you are not sure, read the appropriate sections before making your selection.

☐ Life sciences  ☒ Behavioural & social sciences  ☐ Ecological, evolutionary & environmental sciences

For a reference copy of the document with all sections, see [nature.com/documents/nr-reporting-summary-flat.pdf](nature.com/documents/nr-reporting-summary-flat.pdf)

# Behavioural & social sciences study design

All studies must disclose on these points even when the disclosure is negative.

| | |
|---|---|
| Study description | This studies investigated the relationship between web-browsing behavior and well-being. Study 1 was a longitudinal study, while Study 2 was a cross-sectional study. Studies 3-4 were between-groups studies The data is analysed using quantitative methods. |
| Research sample | Our research sample included participants recruited from Prolific: Study 1 (N = 289, age = 33.17, SD =11.71; females = 50.5%, males = 48.1%, other = 1.4%), Study 2 (N = 447, age = 33.85, SD =12.58; females = 56.4%, males = 41.8%, other = 1.8% ), Study 3 (negative valence condition: N = 55, age=33.96, SD=9.68; females=45.5%, males = 49.1%, other = 5.5%; control condition:N = 47,  age=34.72, SD=12.14; females=46.8%, males = 51.1%, other = 2.1%), Study 4a (label condition: N = 55; no label condition: N = 54, Study 4b (IN = 200, age = 40.8, SD = 12.9; females = 58.0%, males = 50.5%, other, 1.5%).  This allowed us to recruit a large random sample of participants. Our sample is not necessarily representative of the general population. |
| Sampling strategy | For studies 1 and 2, sample size was calculated based on a pilot study. <br> For studies 3 and 4a&b, sample size was checked post-hoc for power. <br> Power analysis for studies 1-4 were conducted using G*power (http://www.psychologie.hhu.de/arbeitsgruppen/ allgemeinepsychologie-und-arbeitspsychologie/gpower.html), with 1-beta = .80 and alpha = 0.05. <br> All studies implemented a convenient sampling strategy (online participants). |
| Data collection | All data was collected online using Qualtrics' survey platform by Christopher Kelly. All data was anonymised. |
| Timing | Study 1: 28/11/2020 - 26/02/2021 <br> Study 2: 22/03/2021 - 11/3/2022 <br> Study 3: 15/08/2022 <br> Study 4a: 22/11/2022 <br> Study 4b: 14/11/2023 |
| Data exclusions | Study 1: 23 participants from whom we could not obtain at least 1KB of text from a minimum of 3 webpages a day were not analyzed. <br> Study 2: 53 participants from whom we could not obtain at least 1KB of text from a minimum of 3 webpages a day were not analyzed. <br> Study 3: 37 participants from whom we could not obtain at least 1KB of text from a minimum of 3 webpages a day were not analyzed. <br> Study 4a: No participants were excluded. <br> Study 4b: No participants were excluded. |
| Non-participation | No participants dropped out/declined participation. |
| Randomization | There are no groups/conditions to be randomized in Studies 1,2,4b. In Studies 3 and 4a, participants were not randomly assigned to different conditions; instead, they unknowingly self-assigned, with conditions conducted within 35-minutes (Study 1) and 120-minutes (Study 2) of each other. The study instructions were identical across the respective study conditions, ensuring participant were not aware to which conditions they were signing up to, nor that there was conditions. |

# Reporting for specific materials, systems and methods

We require information from authors about some types of materials, experimental systems and methods used in many studies. Here, indicate whether each material, system or method listed is relevant to your study. If you are not sure if a list item applies to your research, read the appropriate section before selecting a response.

## Materials & experimental systems

| n/a | Involved in the study |
|-----|----------------------|
| ☒ ☐ | Antibodies |
| ☒ ☐ | Eukaryotic cell lines |
| ☒ ☐ | Palaeontology and archaeology |
| ☒ ☐ | Animals and other organisms |
| ☒ ☐ | Clinical data |
| ☒ ☐ | Dual use research of concern |
| ☒ ☐ | Plants |

## Methods

| n/a | Involved in the study |
|-----|----------------------|
| ☒ ☐ | ChIP-seq |
| ☒ ☐ | Flow cytometry |
| ☒ ☐ | MRI-based neuroimaging |

