## [Peer Review File · Nature Human Behaviour]

Web-Browsing Patterns Reflect and Shape Mood and Mental Health

Corresponding Author: Mr Christopher Kelly

Version 0:

Decision Letter:

16th August 2023

Dear Mr Kelly,

My sincere apologies for the delay of reaching a decision for your manuscript, entitled "Knowledge-Seeking Reflects and Shapes Well-Being", and for your patience during the peer review process.

Your article has now been evaluated by 3 referees. Please note that a fourth reviewer agreed to review, but has not yet returned their comments; we will forward you these independently in the event they are submitted. You will see from our 3 reviewers' comments copied below that, although they find your work of potential interest, they have raised quite substantial concerns. In light of these comments, we cannot accept the manuscript for publication, but would be interested in considering a revised version if you are willing and able to fully address reviewer and editorial concerns.

We hope you will find the referees' comments useful as you decide how to proceed. If you wish to submit a substantially revised manuscript, please bear in mind that we will be reluctant to approach the referees again in the absence of major revisions. We are committed to providing a fair and constructive peer-review process. Do not hesitate to contact us if there are specific requests from the reviewers that you believe are technically impossible or unlikely to yield a meaningful outcome.

To guide the scope of the revisions, the editors discuss the referee reports in detail within the team, including with the chief editor, with a view to (1) identifying key priorities that should be addressed in revision and (2) overruling referee requests that are deemed beyond the scope of the current study. We hope that you will find the prioritised set of referee points to be useful when revising your study. Please do not hesitate to get in touch if you would like to discuss these issues further.

In particular, your revision must address the following (as well as all other reviewer comments):

1) Reviewer 3 raises concerns regarding the novelty of your work, in light of existing related literature. This concern is also echoed by Reviewer 1, who highlights that the current work needs to be better situated in the existing literature. In light of this feedback, in addition to explaining the contribution of your work to the mental health and online activity literature vis-a-vis what is currently known, we ask that you conduct an additional experiment to demonstrate the intervention's effectiveness on well-being (recommended by Reviewer 3).

2) Reviewers 2 and 3 both raise questions regarding the content extracted from web-browsing history, either because the content cannot be retrieved (in the case of social media) or because the content changes quickly. These are fundamental concerns that question the validity of your conclusions. It is therefore important in revision to provide evidence that compellingly addresses these concerns.

3) Reviewer 1 and 2 request a clarification and improvement in terminology use, which needs to be consistently measured across studies.

4) Please report and interpret results between web browsing and smartphone use.

If you wish to submit a suitably revised manuscript, we would hope to receive it within 6 months. I would be grateful if you could contact us as soon as possible if you foresee difficulties with meeting this target resubmission date.

- Include a “Response to the editors and reviewers” document detailing, point-by-point, how you addressed each editor and referee comment. If no action was taken to address a point, you must provide a compelling argument. When formatting this document, please respond to each reviewer comment individually, including the full text of the reviewer comment verbatim followed by your response to the individual point. This response will be used by the editors to evaluate your revision and sent back to the reviewers along with the revised manuscript.
- Highlight all changes made to your manuscript or provide us with a version that tracks changes.

Link Redacted

Thank you for the opportunity to review your work. Please do not hesitate to contact me if you have any questions or would like to discuss the required revisions further.

[redacted]

Reviewer expertise:

Reviewer #1: social media, smartphone use, mental health

Reviewer #2: digital media, social media, psychological wellbeing

Reviewer #3: computational social science, sentiment analysis

REVIEWER COMMENTS:

Reviewer #1:

Remarks to the Author:

I read with great interest the study “Knowledge-Seeking Reflects and Shapes Well-Being”. The paper includes 4 studies, in which the authors explored the relationship between information seeking and well-being in adults. The studies include two exploratory studies, one experiment, and one intervention.

In general, I liked the flow of the paper and the clear writing, although I felt that some key information and theoretical considerations were missing. My general comments are the following:

- I liked the approach used, however, I think that a clear explanation why 4 studies is necessary at the beginning. What is the rationale? Why Study 1+2 (I know that it is for replication but that was not clear in the introduction), why you aimed to give such an overview, how this approach can make the field grow?
- the strength of the paper is the quantification of the affective properties of web pages through machine learning and NLP. That should be emphasized throughout the paper. Why such an approach is important today? What is the potential? How can it foster collaboration among different experts?
- I found the paper short respect to theory, both regarding information seeking and well-being. The authors should stick to the same term (mental health or ill-being), because they did not measure “well”- being. There is an ongoing complex discussion on how to measure well-being and its definition. So, please, be careful and consistent and state clearly what you measured and why.
- Park, C. L., Kubzansky, L. D., Chafouleas, S. M., Davidson, R. J., Keltner, D., Parsafar, P., ... & Wang, K. H. (2023). Emotional well-being: What it is and why it matters. *Affective Science*, 4(1), 10-20.
- Ryan, R. M., & Deci, E. L. (2001). On happiness and human potentials: A review of research on hedonic and eudaimonic well-being. *Annual review of psychology*, 52(1), 141-166.
- also, the paper should include more literature on how emotions drive decision-making processes like information seeking behaviors.
- Lerner, J. S., Li, Y., Valdesolo, P., & Kassam, K. S. (2015). Emotion and decision making. *Annual review of psychology*, 66, 799-823.)
- Hockey, G. R. J., John Maule, A., Clough, P. J., & Bdzola, L. (2000). Effects of negative mood states on risk in everyday decision making. *Cognition & Emotion*, 14(6), 823-855.
- George, J. M., & Dane, E. (2016). Affect, emotion, and decision making. *Organizational Behavior and Human Decision Processes*, 136, 47-55.
- Paulus, M. P., & Angela, J. Y. (2012). Emotion and decision-making: affect-driven belief systems in anxiety and depression. *Trends in cognitive sciences*, 16(9), 476-483.
- Phelps, E. A., Lempert, K. M., & Sokol-Hessner, P. (2014). Emotion and decision making: multiple modulatory neural circuits. *Annual review of neuroscience*, 37, 263-287.
- Finally, I found that many details in the Methods section were missing. How the sample were recruited in each study? From which country? Please report a table with participants’ characteristics for each study (age, gender, SES, education, country, etc), did you randomized? How did you check randomization?
- also, the analyses should be controlled for other variables depending on the randomization procedure
- did you run power analyses for the sample size? What is the effect size?
- Report your limitations and acknowledge which biases you might have introduced (e.g., in Study 3 there is no comparison with

positive emotions).

- Also, you never mentioned the content of browsing, but just the valence. There is a huge gap here on how content could affect the final results. More information in the methods and limitations on that is needed.
- When you suggest possible intervention, you should also acknowledge and cite existing studies looking at interventions.

More detailed comments:

Abstract:

- avoid general statements like "Humans spend many hours searching for knowledge online."
- please clarify your research methodology and rationale

introduction

- you need to define well-being, there are many definitions. Also the difference between ill-being and well-being
- cite more recent literature on the importance of well-being
- "This will inform the development of online tools to enhance well-being and could provide real-time assessment of it" please clarify what do you mean. Real-time assessment of well-being already exists (e.g., digital phenotyping)
- Please, define knowledge-seeking
- "These variations may provide important clues about an individual's inner cognitive and affective state". Add reference
- "In particular, we have theorized that the affective properties of the knowledge people consume from self-guided searches may reflect their well-being (Sharot & Sunstein, 2020)." I think you should contextualize this statement in a more broad perspective on the role of emotions in decision making processes – see suggested citations above.
- "Hence, the relationship between wellbeing and knowledge-seeking may be reciprocal and form a self-reinforcing loop." Please provide citations
- "If indeed a bi-directional relationship exists between the type of information that is consumed from self-guided online searches and well-being, it would have significant theoretical and practical implications. As humans constantly engage in knowledge-seeking online, there is a unique opportunity to harness this data to detect mental health issues and help guide information-seeking patterns. Given this rich potential, it is surprising how limited our knowledge is of the links between well-being and the affective properties information browsed online." I think this statement is too broad. As humans we were always looking for information in books, newspaper, and printed material as well. why searching online would be more impactful or harmful than other sources? Please explain the mechanisms
- What you refer to "psychological and emotional subjective wellbeing" it is not clear. Also, I would suggest to report WHO definition of well-being as well
- I would clearly justify the choice of your research methods to answer your RQ. Why did you run 4 studies and choose this methodology?
- You should also cite more literature on the assessment of emotions through social media content (e.g., many studies looked at Twitter data and mental health for example)
 - Guntuku, S. C., Sherman, G., Stokes, D. C., Agarwal, A. K., Seltzer, E., Merchant, R. M., & Ungar, L. H. (2020). Tracking mental health and symptom mentions on Twitter during COVID-19. *Journal of general internal medicine*, 35, 2798-2800.
 - Valdez, D., Ten Thij, M., Bathina, K., Rutter, L. A., & Bollen, J. (2020). Social media insights into US mental health during the COVID-19 pandemic: Longitudinal analysis of Twitter data. *Journal of medical Internet research*, 22(12), e21418.
- "Participants in Study 1 (N = 289) browsed the web for 20-30 minutes a day for five days, and in Study 2 (N = 447) for 30 minutes on one day." Why two studies with different methodologies and samples? Please, explain
- "Thus, the valence scores of the whole text of a webpage is a good indicator of the valence of text that participants attend to the most". This mechanism is related to attention processes. Eye tracking measures attention (latency) for different part of the webpage. You should interpret your results in light of this.
- You talked about well-being but then quantified mental ill-being (not psychological or emotional well-being). The measure you used it is not very familiar to me, you could have chosen a DSM-5 measure including different symptoms and cut-off criteria. I would better explain why you focused only on Anxious-
 - Depression', 'Social-Withdrawal' and 'Compulsive-Behaviour and Intrusive Thought' (Gillian et al., 2016), why you prioritized these dimensions and what they can predict.
 - You should control your analyses for other covariates, such as education, race, personality, at least in your analyses (especially the regression analysis)
 - The scale to test mood is very simplistic and it refers to happiness (on scales from -50 (very unhappy) to +50 (very happy)), please clarify your choice
 - Why did you study the relationship Knowledge-Seeking and Mood only for negative information? Please, justify your choice. Without testing positive emotions your results might have a huge bias!!
- I do not fully understand the rationale of your intervention. Why would providing people with cues about the valence of webpages alter their knowledge-seeking patterns? Do you mean with respect to the same topic?
- Saying "three Google search result pages" is very broad. Are they controlled for topic? usually one might infer from the topic also the positive/negative content, how you did control for that?
- I think that while you are stating this "Exposure to negative content was in turn associated with worse mood (controlling for pre-browsing mood). We established the causality of this relationship by exposing participants to either negative or neutral webpages." You should also acknowledge the limitations of your work, like that you did not expose participants to positive content, and which the implications might be (e.g., it is likely that the experimental design introduced a bias in the results since you lack the comparison with positive emotions, and exposure with neutral is likely not going to increase mood)
- Also, this is a strong statement "We found that participants' tendency to browse webpages of specific valence, either more negative or positive, was moderately stable over time, indicating that web-browsing behavior has both trait-like and state-like characteristics" and you just used ICCs to sustain this statement. So I would tone down that and again, explicitly consider limitations and what other data (e.g., longitudinal data, EMA) can offer.
- I think that the clear strength of the article is the analysis of the affective properties of text. I would encourage the authors to describe more clearly in both intro and discussion why such an approach is now needed, which are the theoretical and practical

improvements, challenges, and how we can ameliorate such an approach.

- Again, in the discussion you should refer your findings on "providing individuals with cues about the valence of webpages effectively changed their browsing patterns, leading to a decrease in exposure to negative content and an increase in exposure to positive content" to decision-making processes guided by emotions and related that more to existing theories that can explain your results

Methods

- Do you have other information on participants? SES, education level, race, country, etc
- The scales used to assess mental health (or mental ill-being) and the question on the happiness are unrelated theoretically to each other (happiness was not present in the baseline). Also, happiness refers to hedonic well-being, but only ill-being was comprehensively assessed. Be aware of the conceptual difference. Please, justify your choice.
- More control variables would be needed in the ANOVA.
- In study 3 and 4, were participants randomized to the conditions? How did you check randomization?

Reviewer #2:

Remarks to the Author:

Dear Authors,

I was delighted to review the submitted manuscript "Knowledge-Seeking Reflects and Shapes Well-being".

The manuscript reports on four related studies to investigate the (causal) link between the valence of browsed websites and affective well-being, and whether browsing activities can be manipulated by cues on the valence of website content. The authors found that exposure to negatively valenced online content is significantly linked to negative mood, and this relationship is bi-directional. Furthermore, providing labels on the valence of the website next to the links to the websites in a search engine promotes a more frequent selection of positively valenced and less negatively valenced website content.

The major strengths of this manuscript are 1) the use of different methodologies (observational and experimental), 2) the combination of self-reported well-being with browsed website content, on which automated sentiment analysis was performed, and 3) the multiple robustness checks conducted by the authors on the tools used for sentiment analysis, the samples of online content (whole website vs. random sections of a website; eye tracking), and the samples of study participants (participants with submitted browsing history during study period vs. those with submitted browsing history before study period).

At the same time, I have several comments and questions that the authors should consider in a revised version of their manuscript:

1) Concerning the terminology used by the authors, I suggest substituting "knowledge-seeking" with the more general term "information-seeking". One can go even further and use the term "online browsing of web content" as the author did not limit the types of websites consumed by the study participants, and they may provide very different content from news to social support. Since only the valence, not the type of content, was considered in the studies, "online browsing of web content" is more coherent with what participants actually did online.

2) Concerning the ecological validity of the findings, the authors should describe and discuss if they only considered web content browsed on a desktop or also mobile. Since people primarily go online via mobile devices these days, it is key to know if browsing behavior changes due to the mobile version of websites (e.g., How does attention change? How does the scrolling on a website change when done on the mobile screen compared to a desktop device?)

3) If I understood it correctly, the authors used the browsing history of participants (i.e., the list of accessed websites) to access them at a later stage and extract all the text data for sentiment analysis. However, website content may change quickly, especially when considering news websites. Though the authors conducted a robustness check by comparing participants with browsed websites before the actual data collection (including self-reports) and browsed websites during the actual data collection, and found no significant differences, I still wonder if exposure to primarily negatively or positively valenced web content is indeed stable and to what extent it is determined by the algorithm of search engines. In other words, is website browsing user-driven (see selective exposure theories) or more likely provider-driven, or a combination of both? In this regard, I was also wondering if the website order in Study 4 was randomized to rule out order effects in the label condition.

4) This brings me to another point concerning the perceived website valence. While preliminary checks identified a strong correlation between different procedures to evaluate the valence of web content for negatively valenced websites, the ICC for positively valenced websites was only moderate (see page 4. ICC = 0.499). Furthermore, the study findings were rather consistent and robust for negatively valenced web content and negative mood, but non-significant or only partly significant for positively valenced web content and positive mood. The differential effects have not been discussed sufficiently by the authors. Do they think this is a methodological artifact? Or an indicator of the complexity of positive mood/perceptions of positive content? Or maybe exposure time to positively valenced web content was not sufficient to detect a positive bi-directional effect on mood (see assumptions of a dose-response relationship)?

5) In addition to paying more attention to the findings on positive mood and positively valenced websites, it would also be interesting to learn more about the associations between well-being assessed on the three dimensions "Anxious-Depression", "Social-Withdrawal", and "Compulsive-Behaviour and Intrusive Thought" and website browsing. The authors only report on the overall well-being score across the three dimensions, but it would be interesting to learn if the relationships are larger for specific sub-dimensions, e.g., "Anxious-Depression", especially considering that – as reported in the supplement – those with poor mental

health (across the three dimensions) were more exposed to fear-related websites.

In addition to these 5 comments, I also have some minor suggestions regarding the structure/writing of the manuscript itself:

- 1) I suggest integrating the last paragraph of the results section in the discussion to avoid repetition.
- 2) In the discussion on p. 11, last paragraph, the authors should add some references to their statement "Previous research in this area has primarily focused on ..."
- 3) There are some minor grammar mistakes and typos that should be checked and resolved, e.g., in the abstract properties influence/alter instead of influences/alters; p. 5 NRC VAD lecion should be NRC VAD lexicon; p. 19 first sentence no comma before "knowledge?"
- 4) Formatting of the title on p. 7, no " _ "

Reviewer #3:

Remarks to the Author:

Summary:

This paper aims to examine how online knowledge-seeking affects well-being. Through four studies involving 947 participants, the researchers explore the connection between the affective properties of online information and individuals' psychological and emotional well-being. The results indicate that engaging with negative information online is associated with worse mental health and mood. By manipulating variables in the experimental setting, the researchers establish a causal reciprocal relationship between affective characteristics and well-being. Moreover, they devise an intervention that effectively modifies web-browsing habits to be less negative. These findings suggest that the type of knowledge individuals seek online reflects and influences their mental state, establishing a feedback loop that may contribute to ongoing mental health challenges. The study also proposes a potential approach for evaluating and improving welfare in the digital era by acknowledging the emotional impact of web content.

The paper is well-written and clearly structured. However, I have major concerns regarding the contribution and methodology of this work. Hence, I cannot recommend acceptance of this paper. Below I have listed some comments that may help the authors for future submissions of the paper.

Major Comments:

1. Novelty and positioning

A major concern with this paper is the very shallow discussion of related work. Neither in the Introduction nor in the Discussion section do the authors engage with the literature on mental health and online activity. As such, it is impossible to evaluate the contribution of this work over others that study online information-seeking (e.g., related to Covid, social networking sites, etc.) and mental health outcomes [1, 2, 3, 4, 5, 6]. Given that the paper does not disclose how it is situated in the existing literature, I have reservations as to what contribution the paper provides and whether it significantly advances our understanding of mental health and online activities.

2. Methodological questions

The authors state that they recruit participants from Prolific, but where the specific inclusion criteria? Did participants have to speak English? Was the recruitment restricted to a specific geographic entity (e.g., US) or worldwide? In general, the authors should add more information on how participants were selected. For example, the authors should clearly state their inclusion or exclusion criteria. This information is vital for assessing the generalizability of the findings and ensuring the representativeness of the sample.

How was the sample size determined? I did not find a preregistration of that (which I would have loved to see). Compared to other studies testing e.g., behavioral interventions, I personally find the sample size somewhat small.

The paper states that users had to browse the web via Firefox. How was this asserted? Does this bias the results (e.g., to tech-savvy population)? I am surprised that this is not discussed as part of the limitations.

How were social media pages handled? Note that the authors may get the link to the social media page as part of the browsing history but, due to security mechanisms, cannot crawl the actual content that is displayed. Hence, the "private" link will resolve to the landing page (e.g., facebook.com). I think this misses a major part of the actual online behavior. More crucially, this limitation is neither properly disclosed nor discussed (and not tested for as a robustness check). On the contrary, it may even lead to bias. As a thought experiment, let the landing page of facebook.com only contain negative words, then this may confound the analysis, so that the authors find that users with more social media consumption also experience lower well-being, even though this had little to nothing to do with the negative words (but maybe other parts on the social media page such as peer pressure etc.).

Figure 2 refers to data from N=10 participants. However, the point plot has way more than 10 points. Something is odd here (either the N is not correct, or the authors have multiple observations per person, which would imply that a different analysis is needed). Likewise, it is unclear what confidence intervals are shown. 95%?

The paper mentions that participants were instructed to browse the internet for studies 1 and 2 but are calm about further details

(e.g. was there a suggested starting point). It is essential to clarify the specific guidelines given to participants to ensure consistency across the experimental conditions. In addition, I was wondering how the participants were introduced and debriefed to the experiment.

I would like to see more summary stats. As of now, the paper is mostly focused on the significance testing but some descriptives on the survey part, the sentiment variables, etc. would be helpful (e.g., I would like to see the mood variable distribution). It would also be interesting to see some examples of websites visited by the participants during the experiment (e.g., a table of the top 10 URLs). Did the authors allow individuals to participate in multiple studies or did they limit participation to just one study? Which software was used for the analysis?

A minor point: did the participants give informed consent. While this is not a big issue, I would recommend a statement, even if it is just a sub-clause.

3. Generalizability in the field

The paper provides a host of evidence from the lab. However, the authors do not discuss how their results transfer to the field. Actively asking participants to browse may change their state of mind (e.g., make users adopt a utilitarian mindset instead of a hedonic mindset) and thus affect how they perceive web pages and what they search for. I would like to see additional explanations as to why the measured effect is transferable to the field. As an idea, instead of only checking the valence of the websites, the authors could, for example, check the frequency of URLs.

Relatedly, the authors let participants "browse" two web pages in the first part of study 3. Does this really reflect a typical browsing activity where I would assume that people randomly visit multiple websites? Could it be that the effect measured for mood is solely due to the negative content rather than the activity itself?

The authors are also calm about the devices used during the experiment. I guess that many people would nowadays browse with mobile devices. Did all participants use the same type of device when conducting the experiment? If not, have the authors checked how effects vary between mobile and stationary devices? This could give several interesting analyses for the supplements.

4. Questionable labeling in Study 4

I am wondering why the authors chose to label the intervention "feel better", "feel worse" and "neutral". These labels seem to indicate an action associated with the choice of a certain webpage rather than an assessment of the web page's valence. I am concerned that this might have introduced a bias in the choice by participants since it seems unlikely somebody is going to click on a webpage labeled as "feel worse."

A crucial extension of study 4 would be to examine the impact of the intervention on participants' well-being. While the study demonstrates the intervention's positive influence on browsing behavior, it remains unclear how this intervention affects the resulting change in well-being. Exploring this aspect would provide a comprehensive understanding of the intervention's effectiveness. As a case in point, such assessment is crucial to understand the effect size of the intervention and to judge whether it is meaningful in practice.

5. Measures and methodological approach

For the calculation of Psychopathology Dimension Scores (in studies 1 and 2), clarification is needed on what these terms entail and how they are measured. In addition, it is unclear which questionnaire results (pre- or post-web-browsing) were used to calculate the three psychopathology dimension scores. This detail is crucial for understanding the associations between web browsing and psychopathology.

I am further wondering why the authors chose to include the psychopathology features in their analysis. At the moment, they are not included consistently in the study. The authors measure them in studies 1 and 2 but chose not to intervene on them in studies 3 and 4. Why are the psychopathology features not intervened in studies 3 and 4?

The authors chose self-reported "Mood" as a second dependent variable measured on a scale from -50 to + 50. The authors should provide additional information on why they chose this scale and how reliable it is. Relatedly, how did the authors set the endpoints and how does this influence the results? Could this scale be too fine-grained for participants (e.g., what is the difference between 30 and 31 but also between 30 and 40)?

I am struggling with the consistency of the methods. In some places, the authors employ linear regression analysis but ANOVA in others. This makes it hard to compare the results. To streamline the methodological approach for the studies, I strongly advise employing a regression analysis approach rather than comparing two groups using ANOVA in study 3. In the regression equation, the valence score (ranging from negative to positive) of web pages could be incorporated as a single continuous variable. This would enable the interpretation of the coefficient of this variable in a causal manner. Moreover, considering the change in well-being (post vs. pre-browsing) as the dependent variable would provide more valuable insights and easier interpretation. Similarly, for study 4, I would recommend utilizing a regression analysis approach, treating the variable of interest as the treatment assignment to the label condition versus the no-label condition. (As a result of the regression analyses, they could also run an additional check where they control for browsing time, as the difference in browsing time may explain also differences in the dependent variable.)

Also, I suggest the authors streamline their usage of dependent variables to one per category. For example for the mood as the dependent variable, I would highly suggest the authors either always use (a) the delta in the well-being (i.e., post-browsing mood - pre-browsing mood) or (b) the post-browsing mood and a control for the pre-browsing mood. This would greatly help the comparability of the different studies.

An important part of well-being is positivity. The authors have analyzed this in studies 1 and 2 but not in study 3. I was wondering why. It seems like it would have been natural to check how browsing positive websites affects mood and whether a positive mood leads to positive browsing as indicated in studies 1 and 2. Why have the authors omitted this in study 3? (I think that this, in general, could be a nice addition for a supplement.)

6. Clearer discussion of limitations and findings

Moreover, a more detailed discussion of the limitations of the study design is needed. For example, participants might have chosen to browse only certain web pages or cleaned their search history post-hoc since they knew it would be viewed. In addition, self-reported emotions might be biased as well.

How do the findings add to existing research? Others have found that negative emotions on social media are contagious? Are the authors effect sizes of similar magnitude? In general, I would like to see more discuss of the effect size and whether they are meaningful? (as of now, the authors appear to primarily search for "significance" but with little focus on the actual impact).
Validation of experimental setup

7. Lexicon-based Emotion Detection

The authors validate the use of the NRC lexicon for quantifying the affective properties of web pages by (a) comparing it to other dictionary-based as well as machine-learning methods and by (b) validating it using a comparison to human annotation, which is commendable. However, information about the interclass correlation coefficient (ICC) between the machine learning approach (Distilbert) and human assessment would provide more justification and validity in using the NRC lexicon, as human annotation should be regarded as the gold standard. Also, it would be beneficial to elaborate on the potential limitations of the selected lexicon.

Why was the ICC only computed for a subset of 100 participants? How was the sentiment variable aggregated over different pages (e.g., sum, average)?

8. Minor Comments:

Reliability of Browsing History Submission: The paper states that participants were asked to confirm the accuracy of their browsing history submissions. As an alternative, one could consider other methods, such as checking timestamps within the browsing history, to verify the validity of the submissions.

The authors might want to consider restructuring parts of the paper. On the one hand, I would suggest removing the validation studies from the results section and instead adding a subsection "validation" to the methodology section. On the other hand, authors might want to add tables with the results from the regression and/or ANOVA analysis to provide a clearer overview.

References

- [1] Charpentier, C. J., Cogliati Dezza, I., Vellani, V., Globig, L. K., Gädeke, M., & Sharot, T. (2022). Anxiety increases information-seeking in response to large changes. *Scientific Reports*, 12(1), 7385.
- [2] Braghieri, L., Levy, R. E., & Makarin, A. (2022). Social media and mental health. *American Economic Review*, 112(11), 3660-3693.
- [3] Wilding, S., O'Connor, D. B., Ferguson, E., Wetherall, K., Cleare, S., O'Carroll, R. E., ... & O'Connor, R. C. (2022). Information seeking, mental health and loneliness: Longitudinal analyses of adults in the UK COVID-19 mental health and wellbeing study. *Psychiatry research*, 317, 114876.
- [4] Toseeb, U., & Inkster, B. (2015). Online social networking sites and mental health research. *Frontiers in psychiatry*, 6, 36.
- [5] Bell, V. (2007). Online information, extreme communities and internet therapy: Is the internet good for our mental health?. *Journal of mental health*, 16(4), 445-457.
- [6] Frison, E., & Eggermont, S. (2017). Browsing, posting, and liking on Instagram: The reciprocal relationships between different types of Instagram use and adolescents' depressed mood. *Cyberpsychology, Behavior, and Social Networking*, 20(10), 603-609.

Version 1:

Decision Letter:

Our ref: NATHUMBEHAV-23041322A

3rd July 2024

Dear Dr. Kelly,

Thank you for submitting your revised manuscript "Web-Browsing Patterns Reflect and Shape Mental Health" (NATHUMBEHAV-23041322A). It has now been seen by the original referees and their comments are below. As you can see, the reviewers find that the paper has improved in revision. We will therefore be happy in principle to publish it in Nature Human Behaviour, pending minor revisions to satisfy the referees' final requests and to comply with our editorial and formatting guidelines.

We are now performing detailed checks on your paper and will send you a checklist detailing our editorial and formatting requirements within two weeks. Please do not upload the final materials and make any revisions until you receive this additional information from us.

[redacted]

Reviewer #1 (Remarks to the Author):

The authors did a great job in ameliorating the manuscript, which I think it is now ready for publication

Reviewer #2 (Remarks to the Author):

Dear Editor,
Dear Authors,

Many thanks for the revised version of this manuscript now entitled "Web-Browsing Patterns Reflect and Shape Mental Health".

I have read with great interest all the reviewers' comments and the authors' responses to them. The provided rationales for both changes to the manuscript and kept sections are sound. I particularly appreciate the additional validation studies conducted on the basis of my and the other reviewers' comments (e.g., browsing behaviour in different devices, stability of website valence).

To further improve the manuscript, I would like to echo a comment made by the other reviewers concerning the conceptualization and terminology of mental health. Only Study 2 considered mental health with its different dimensions, while Studies 3 and 4 b considered people's mood. The authors write on page 8 that "Next, we ask whether it is also associated with mood, which is a feature of mental health, and if so whether this association is bidirectional." Here, they should be more specific and include a more detailed description and literature supporting their claim how mood can be considered a feature of mental health.

A minor and final comment regards the citation style and reference list, with citations not being consistent (e.g., use of & and "and") and the reference list not alphabetically ordered or numbered, based on the style that the journal requires.

Sincerely,
Anne-Linda Camerini

Reviewer #3 (Remarks to the Author):

Second round review: Web-Browsing Patterns Reflect and Shape Mental Health

I thank the authors for answering my questions and am generally satisfied with the responses to my comments. Overall, the manuscript has much improved. Below, I have a few comments that I recommend to be addressed:

Practical implications:

I would like to hear more about how the authors envision their intervention set into practice. I fully agree that valence should not be the main factor to guide information-seeking. Yet, I find it important to provide more about how to put this into practice. For example, what other labels would be needed, how scalable is the intervention, who would be the best target group, and who would adopt it?

Recruitment of study 1:

I recommend to include another robustness check on how omitting the 59 participants not from the US or UK affects the results.

Validation of sentiment measures:

Please add details on how the output of DistillBERT was mapped to sentiment scores.

Please add the exact scale on which sentiment was measured to compare the valence of web pages across devices. I could not find the Likert scale mentioned in the main file in the supplements.

Website visits:

I understand the concerns regarding the privacy of the participants. A solution may be to provide the top 5/10 main URLs (e.g. <https://www.nytimes.com/>) aggregated across participants. Thereby, one would preserve the privacy of users and provide readers with a better idea of typical browsing activities during the experiment.

Reproducibility:

I checked the Github shared in the "Data and Code availability" statement. I believe this is not all materials necessary to replicate the results. Also, a ReadMe with more information on how to execute the Code including short file descriptions would be necessary to ensure reproducibility.

I also recommend include the specific R packages used for the data analysis.

Minor:

It would be helpful to link study instructions in the supplements separately for each study.

I would encourage to include a short summary of the results of the additional studies regarding device sensitivity in the main document.

"[...] who observed the labels would chose to exposed themselves [...]" => "expose themselves"

Again, I am convinced that the paper will spur an extensive discussion and interesting follow-up research.

Signed: Stefan Feuerriegel (with help of team members; the senior member is solely responsible for errors in the above reviewer).

Version 2:

Decision Letter:

Dear Mr Kelly,

We are pleased to inform you that your Article "Web-Browsing Patterns Reflect and Shape Mood and Mental Health", has now been accepted for publication in *Nature Human Behaviour*.

Please note that *Nature Human Behaviour* is a Transformative Journal (TJ). Authors may publish their research with us through the traditional subscription access route or make their paper immediately open access through payment of an article-processing charge (APC). Authors will not be required to make a final decision about access to their article until it has been accepted. [Find out more about Transformative Journals](https://www.springernature.com/gp/open-research/transformative-journals)

Authors may need to take specific actions to achieve [compliance with funder and institutional open access mandates](https://www.springernature.com/gp/open-research/funding/policy-compliance-faqs). If your research is supported by a funder that requires immediate open access (e.g. according to [Plan S principles](https://www.springernature.com/gp/open-research/plan-s-compliance)) then you should select the gold OA route, and we will direct you to the compliant route where possible. For authors selecting the subscription publication route, the journal's standard licensing terms will need to be accepted, including [self-archiving policies](https://www.springernature.com/gp/open-research/policies/journal-policies). Those licensing terms will supersede any other terms that the author or any third party may assert apply to any version of the manuscript.

We welcome the submission of potential cover material (including a short caption of around 40 words) related to your manuscript; suggestions should be sent to *Nature Human Behaviour* as electronic files (the image should be 300 dpi at 210 x 297 mm in either TIFF or JPEG format). Please note that such pictures should be selected more for their aesthetic appeal than for their scientific

content, and that colour images work better than black and white or grayscale images. Please do not try to design a cover with the Nature Human Behaviour logo etc., and please do not submit composites of images related to your work. I am sure you will understand that we cannot make any promise as to whether any of your suggestions might be selected for the cover of the journal.

[redacted]

P.S. Click on the following link if you would like to recommend Nature Human Behaviour to your librarian
<http://www.nature.com/subscriptions/recommend.html#forms>

** Visit the Springer Nature Editorial and Publishing website at http://editorial-jobs.springernature.com?utm_source=ejp_NHumB_email&utm_medium=ejp_NHumB_email&utm_campaign=ejp_NHumB for more information about our career opportunities. If you have any questions please click [here](mailto:editorial.publishing.jobs@springernature.com).

Open Access This Peer Review File is licensed under a Creative Commons Attribution 4.0 International License, which permits use, sharing, adaptation, distribution and reproduction in any medium or format, as long as you give appropriate credit to the original author(s) and the source, provide a link to the Creative Commons license, and indicate if changes were made. In cases where reviewers are anonymous, credit should be given to 'Anonymous Referee' and the source. The images or other third party material in this Peer Review File are included in the article's Creative Commons license, unless indicated otherwise in a credit line to the material. If material is not included in the article's Creative Commons license and your intended use is not permitted by statutory regulation or exceeds the permitted use, you will need to obtain permission directly from the copyright holder.

We would like to thank the Reviewers for a thorough and thoughtful commentary on our work. Incorporating their suggestions has strengthened the manuscript which significantly. To ensure we address all of the Reviewers' comments and for ease of reference, we have included the reviews below (in BOLD), followed by our response to each concern.

Reviewer #1:

Remarks to the Author:

I read with great interest the study “Knowledge-Seeking Reflects and Shapes Well-Being”. The paper includes 4 studies, in which the authors explored the relationship between information seeking and well-being in adults. The studies include two exploratory studies, one experiment, and one intervention. In general, I liked the flow of the paper and the clear writing, although I felt that some key information and theoretical considerations were missing.

- We thank the Reviewer for this positive assessment and address all their comments below.

My general comments are the following:

- I liked the approach used, however, I think that a clear explanation why 4 studies is necessary at the beginning. What is the rationale? Why Study 1+2 (I know that it is for replication but that was not clear in the introduction), why you aimed to give such an overview, how this approach can make the field grow?

- We conducted four interconnected studies to build a comprehensive understanding of the relationship between the valence of content browsed online and mental health. First, Study 1 tests for a correlation between the valence of content browsed and mental health. Study 2 then replicates the results observed in Study 1, to assure that they are robust. Study 3 then tests for a causal relationship between the valence of content browsed and mood. Finally, Study 4 investigated whether an intervention based on these insights can effectively alter web-browsing patterns to improve mood. We now describe the rationale for each study in the Introduction (**pg. 4**). Together, the studies suggest that the relationship between web-browsing patterns and mental health is causal and bi-directional.
- The approach advances the field both theoretically and practically. Theoretically, it shows how mental health is both reflected in the content browsed and is impacted by it. Methodologically, it opens up a new line of research with a technique that is completely novel (analysis of content browsed). The empirical findings also feed into the development of tools that can help users browse the web in an informed manner that can improve mood. Beyond these, the paper highlights the importance of replication, causal testing and bridging theory with practice. We now highlight these points in the Discussion (**pg. 12-14**).

- the strength of the paper is the quantification of the affective properties of web pages through machine learning and NLP. That should be emphasized throughout the paper.

Why such an approach is important today? What is the potential? How can it foster collaboration among different experts?

- We appreciate the Reviewer's comment. We agree that the methodology of our paper is a notable strength and now emphasize it throughout the paper (pg. 3 & 12-14). This approach is especially important today, as individuals constantly browse information online. Thus, it is critical to understand how the content they browse may impact their mental health and what can be done to facilitate or mitigate these effects. Here, we bridge psychological theory with computer science, both to advance theory and inform the development of practical tools. We believe such interdisciplinary approach has great potential. We now highlight these points in the Discussion (pg. 12-14).

- I found the paper short respect to theory, both regarding information seeking and well-being. The authors should stick to the same term (mental health or ill-being), because they did not measure “well”- being. There is an ongoing complex discussion on how to measure well-being and its definition. So, please, be careful and consistent and state clearly what you measured and why.

• Park, C. L., Kubzansky, L. D., Chafouleas, S. M., Davidson, R. J., Keltner, D., Parsafar, P., ... & Wang, K. H. (2023). Emotional well-being: What it is and why it matters. *Affective Science*, 4(1), 10-20.

• Ryan, R. M., & Deci, E. L. (2001). On happiness and human potentials: A review of research on hedonic and eudaimonic well-being. *Annual review of psychology*, 52(1), 141-166.

-also, the paper should include more literature on how emotions drive decision-making processes like information seeking behaviors.

• Lerner, J. S., Li, Y., Valdesolo, P., & Kassam, K. S. (2015). Emotion and decision making. *Annual review of psychology*, 66, 799-823.)

• Hockey, G. R. J., John Maule, A., Clough, P. J., & Bdzola, L. (2000). Effects of negative mood states on risk in everyday decision making. *Cognition & Emotion*, 14(6), 823-855.

• George, J. M., & Dane, E. (2016). Affect, emotion, and decision making. *Organizational Behavior and Human Decision Processes*, 136, 47-55.

• Paulus, M. P., & Angela, J. Y. (2012). Emotion and decision-making: affect-driven belief systems in anxiety and depression. *Trends in cognitive sciences*, 16(9), 476-483.

• Phelps, E. A., Lempert, K. M., & Sokol-Hessner, P. (2014). Emotion and decision making: multiple modulatory neural circuits. *Annual review of neuroscience*, 37, 263-287.

- Following the Reviewer's feedback we now use the term 'mental health' rather than 'well-being.'
- In addition, we now elaborate on the relevant theory that led to our hypothesis (pg. 3): Specifically, we state that “negative thoughts may lead to searches for information with a similar sentiment, resulting in the consumption of negatively valenced content, which could in turn exacerbate one’s negative affective state. This potential mechanism is consistent with findings suggesting that people with depression tend to engage with stimuli that perpetuate their sadness (Milgram et al., 2015), and is analogous to the mechanism hypothesised to underlie rumination. Specifically, it has been suggested that continuous negative thoughts (akin to internal information-seeking) can sustain and exacerbate low moods through a feedback loop (Watkins, 2008; Michl et al., 2013). In a similar vein, an individual's affective state may influence, and be influenced by, the type of information sought from external sources (as hypothesised in Sharot & Sunstein,

2020)” (pg. 3). We also thank the reviewer for the references in the field of decision making. We now mention that the ideas are consistent with the above references in the sense that they show affect alters decision making processes. However, at the end we felt that the references that were most relevant to our precise theory were those that we mentioned in the paragraph above (pg. 3).

- Finally, I found that many details in the Methods section were missing. How the sample were recruited in each study?

- All samples were recruited via Prolific. We now clearly state this in the methods along with our inclusion criteria for each study (pg. 14-18).

From which country?

- Because our study was conducted in English all participants in Studies 2-4 were recruited from the UK and US. In Study 1, due to a technical error 59 out of 349 were from other countries . We do not have a specific numerical breakdown for each specific country. We did, however ask whether English was participants’ first language and this covariate is now added to the analysis and the results remain unchanged (see **Supplementary**).

Please report a table with participants’ characteristics for each study (age, gender, SES, education, country, etc),

- We have now added demographics tables with participants’ characteristics for each study: age, gender, ethnicity, income, education, and first language (see **Supplementary**).

did you randomized?

- In Studies 1 and 2 there are no conditions to be randomized. For Studies 3 and 4 the advertisement for the studies were exactly the same for the different conditions, thus participants could not tell them apart when deciding to participate (pg. 17 & 18). In other words, we did not randomly assign them to different conditions, but rather they unknowingly assigned themselves, with conditions being run approximately only 35min-120min apart (see pg. 17 & 18). As detailed in the response below, our checks confirm no difference between groups on any demographical characteristics (see **Supplementary**).

How did you check randomization?

- To check randomization in self assignment, we assessed whether there were any differences between conditions with regard to demographic and socio-economic variables using a series of independent t-tests and chi-square tests. No significant differences were observed (see **Supplementary**).

also, the analyses should be controlled for other variables depending on the randomization procedure

- We now control for all available demographic and socio-economic variables in all analysis and find that all results remain unchanged (see **Supplementary**).

- did you run power analyses for the sample size? What is the effect size?

- Yes, for all studies we conducted pilot studies to determine the sample size needed to observe the effects of interest. Power analysis was run using the G*Power 3.1 software, with an alpha of 0.05 and a 1-beta of 0.95. We now report details about power analyses in the methods section (pg. 14-19) and report all effect sizes in the main text (pg. 4-12). The different effect sizes are $0.1 \leq r \leq 0.138$; $0.017 \leq \text{partial eta squared} \leq 0.151$; $0.044 \leq \text{Beta} \leq 0.503$; Cohen's $d = 0.436$.

- Report your limitations and acknowledge which biases you might have introduced (e.g., in Study 3 there is no comparison with positive emotions).

- Thank you for prompting us to discuss limitations, which we now do in the discussion section (pg. 13). Our sentiment analysis is exclusively of text and including images and videos would be valuable in future studies. Such studies could also measure the exact amount of time users spend on each piece of content, providing a more precise estimate of the relationship between valence of content consumed and mental health and include an analysis of password-protected websites. It would also be helpful to test for the effects using browsers beyond Firefox. With regards to Study 3, negative valence was compared to neutral as a replicable relationship was observed between browsing negative content and mental health and no such replicable relationship was observed for positive content. This comparison does not introduce any systematic biases or confounds that we are aware of and is a common approach. The aim of Study 3 was to test for causality of the relationships observed in Studies 1 and 2.

- Also, you never mentioned the content of browsing, but just the valence. There is a huge gap here on how content could affect the final results. More information in the methods and limitations on that is needed.

- Thank you for this comment. The explicit aim of the study was to test the relationship between the valence of information consumed from self-guided searches and mental health. The rationale was that (i) negative valence can trigger symptoms and/or make them worse, for example by inducing stress and (ii) searches of negative information may reflect negative thoughts, which are associated with mental health problems. We did not conduct a content analysis as we did not have a specific hypothesis or underlying theory that we aimed to test. However, our new Study 4b shows that valence of webpages selected for consumption is associated with mood change in the hypothesized direction, even when the topic is held constant across valence conditions (pg. 11-12 & 18). This suggests that valence indeed is the critical factor here. While exploring the connection between the topics of webpages browsed and mental health is undoubtedly intriguing, it falls outside the objectives of this particular project.

- When you suggest possible intervention, you should also acknowledge and cite existing studies looking at interventions.

- Thank you for this suggestion, we now cite studies that have tested online interventions to alter web-browsing behaviour such as screentime awareness tools (Kim et al., 2016; Kovacs et al., 2022; see pg. 3).

More detailed comments:

Abstract:

- avoid general statements like “Humans spend many hours searching for knowledge online.”

- We have now removed this general statement in place of an empirical statement: *“Humans spend on average 6.5 hours a day online.”* (see Kemp, 2023).

- please clarify your research methodology and rationale

- We have now clarified our methodology and rationale in the abstract.

Introduction

- you need to define well-being, there are many definitions. Also the difference between ill-being and well-being

- Following the Reviewer's earlier recommendation, we now use the term ‘mental health’ rather than ‘well-being’. The exact definition of mental health differs across countries, scientists and organizations. The CDC states that *“Mental health includes our emotional, psychological, and social well-being”*; MIND – one of the largest mental health charities in the UK – states *“Mental health is a state of mental well-being”*. Almost all organizations, including the UK’s NHS, include in their definition *“mental health conditions”* or *“mental health problems”* listing psychopathology disorders. Thus, under the majority of definitions mental health problems are associated with psychopathology, and mental health with a positive emotional state, which also include mood and happiness. We now discuss this in the Introduction (pg. 3).

- cite more recent literature on the importance of well-being

- We now use the term ‘mental health’ instead of ‘well-being’. Thus, in line with the Reviewer’s comment, we now cite more recent literature on the importance of mental health (pg. 3)

- “This will inform the development of online tools to enhance well-being and could provide real-time assessment of it” please clarify what do you mean. Real-time assessment of well-being already exists (e.g., digital phenotyping)

- We now reword the sentence to make clear that we mean additional real-time assessment methods that can be used together with existing digital phenotyping tools (pg. 3).

- Please, define knowledge-seeking

- We now use the term ‘information-seeking’ in place of ‘knowledge-seeking’, as suggested by Reviewer 2. Information-seeking is the active pursuit of knowledge (Sharot & Sunstein, 2020). We now state this in the introduction (pg. 3).

- “These variations may provide important clues about an individual’s inner cognitive and affective state”. Add reference

- We have now added a reference in support of this statement (pg. 3).

- “In particular, we have theorized that the affective properties of the knowledge people consume from self-guided searches may reflect their well-being (Sharot & Sunstein, 2020).” I think you should contextualize this statement in a more broad perspective on the role of emotions in decision making processes – see suggested citations above.

- Thank you for this suggestion. As the literature on emotions and decision making is vast, to insure the paper’s focus is maintained we opted to elaborate on the literature that most closely relates to our specific hypothesis (pg 3). Thus, while we do now mention that the hypothesis aligns with the vast literature showing affect alters decision making (referencing some of the suggested papers) we elaborate only on those references that directly relates to the predictions (top of pg 3). Specifically, we state that “negative thoughts may lead to searches for information with a similar sentiment, resulting in the consumption of negatively valanced content, which could in turn exacerbate one’s negative affective state. This potential mechanism is consistent with findings suggesting that people with depression tend to engage with stimuli that perpetuate their sadness (Milgram et al., 2015), and is analogous to the mechanism hypothesised to underlie rumination. Specifically, it has been suggested that continuous negative thoughts (akin to internal information-seeking) can sustain and exacerbate low moods through a feedback loop (Watkins, 2008; Michl et al., 2013). In a similar vein, an individual's affective state may influence, and be influenced by, the type of information sought from external sources (as hypothesised in Sharot & Sunstein, 2020)” (pg. 3).

- “Hence, the relationship between wellbeing and knowledge-seeking may be reciprocal and form a self-reinforcing loop.” Please provide citations

- We have now reworded this statement and have added references in support of the hypothesis, which we test in this manuscript (see pg. 3).

- “If indeed a bi-directional relationship exists between the type of information that is consumed from self-guided online searches and well-being, it would have significant theoretical and practical implications. As humans constantly engage in knowledge-seeking online, there is a unique opportunity to harness this data to detect mental health issues and help guide information-seeking patterns. Given this rich potential, it is surprising how limited our knowledge is of the links between well-being and the affective properties information browsed online.” I think this statement is too broad. As humans we were always looking for information in books, newspaper, and printed material as well. Why searching online would be more impactful or harmful than other sources? Please explain the mechanisms

- The reason web-browsing is distinct from other forms of information-seeking is magnitude. The amount of information available to humans online, and the ease and speed by which it can be consumed is many magnitudes greater than before the internet was introduced. Thus, specific information-seeking patterns that already existed, may have a much greater impact on modern day humans. Moreover, it is easier to monitor

and assess online behaviour due to the digital nature of it, thus the practical applications are also greater than before. We have now re-written this section (see pg.3).

- What you refer to “psychological and emotional subjective wellbeing” it is not clear. Also, I would suggest to report WHO definition of well-being as well

- In response to the Reviewer’s earlier comments, we now use the term ‘mental health’ instead of ‘well-being’, and discuss its definition (see pg. 3), as noted in response to a previous comment.

- I would clearly justify the choice of your research methods to answer your RQ. Why did you run 4 studies and choose this methodology?

- We conducted four interconnected studies to build a comprehensive understanding of the relationship between the valence of content browsed online and mental health. First, Study 1 tests for a correlation between the valence of content browsed and mental health. Study 2 then replicates the results observe in Study 1, to assure they are robust. Study 3 then tests for a causal relationship between the valence of content browsed by subjects and mood. Finally, Study 4 investigates whether interventions based on these insights can effectively alter web-browsing patterns to improve mood. Together, the studies suggest that the relationship between web-browsing patterns and mental health is causal and bi-directional. The empirical findings inform the development of tools that may improve web-browsing. We now make this explicit in the Introduction and hope that the paper may encourage the practice of replication, causal testing and bridging theory with practice (pg. 3).

- You should also cite more literature on the assessment of emotions through social media content (e.g., many studies looked at Twitter data and mental health for example)

- Guntuku, S. C., Sherman, G., Stokes, D. C., Agarwal, A. K., Seltzer, E., Merchant, R. M., & Ungar, L. H. (2020). Tracking mental health and symptom mentions on Twitter during COVID-19. *Journal of general internal medicine*, 35, 2798-2800.
- Valdez, D., Ten Thij, M., Bathina, K., Rutter, L. A., & Bollen, J. (2020). Social media insights into US mental health during the COVID-19 pandemic: Longitudinal analysis of Twitter data. *Journal of medical Internet research*, 22(12), e21418.

- We thank the Reviewer for their comment. We now cite the literature assessing emotions via content that is shared on social media in the introduction (pg. 3).

- “Participants in Study 1 (N = 289) browsed the web for 20-30 minutes a day for five days, and in Study 2 (N = 447) for 30 minutes on one day.” Why two studies with different methodologies and samples? Please, explain

- After running Study 1 we examined our data with the aim of balancing statistical robustness and resource efficiency, including financial considerations. Our findings suggested that conducting the study with about 500 participants in a single day could yield substantial cost savings. Specifically, data from Study 1 revealed a high effect size for the relationship between the Negative Score of web-browsing on day one and mental health (G*Power: alpha = 0.05, 1-beta = 0.95). We also opted to extend the data collection period marginally to ensure we yield enough data per participant. We now detail this rational in the revised manuscript (pg. 17).

- “Thus, the valence scores of the whole text of a webpage is a good indicator of the valence of text that participants attend to the most”. This mechanism is related to attention processes. Eye tracking measures attention (latency) for different part of the webpage. You should interpret your results in light of this.

- We thank the Reviewer for their comment. We now interpret the results with respect to attention processes (pg. 5).

- You talked about well-being but then quantified mental ill-being (not psychological or emotional well-being). The measure you used it is not very familiar to me, you could have chosen a DSM-5 measure including different symptoms and cut-off criteria. I would better explain why you focused only on Anxious-Depression’, ‘Social-Withdrawal’ and ‘Compulsive-Behaviour and Intrusive Thought’ (Gillan et al., 2016), why you prioritized these dimensions and what they can predict.

- We thank the Reviewer for prompting us to explain our choice. The dimensionality approach is beneficial over traditional psychopathology classifications as it allows for the identification of symptoms that may be indicative of multiple psychiatric conditions, thereby providing a more nuanced understanding that crosses conventional clinical categories (Gillan et al., 2016; Rouault et al., 2018; Seow & Gillan, 2020; Kelly & Sharot, 2021). This has become a well-established approach in the field (Gillan et al., 2016; Rouault et al., 2018; Seow & Gillan, 2020; Kelly & Sharot, 2021; Cuthbert & Insel, 2010; Cuthbert & Insel, 2013). We now make this clear in the Results section (pg. 6).

- You should control your analyses for other covariates, such as education, race, personality, at least in your analyses (especially the regression analysis)

- Thank you, we now control for income, education, ethnicity, age, gender, and English as first language – the results remain consistent. We report this analysis in the **Supplementary** for Studies 1-4, and mention that results remain consistent throughout the paper (pg. 8, 10, 11, & 12).

- The scale to test mood is very simplistic and it refers to happiness (on scales from -50 (very unhappy) to +50 (very happy), please clarify your choice

- We chose to assess happiness levels as happiness is considered a key component of overall well-being (Medvedev & Landhuis, 2018) and mental health (APA, 2023). Indeed, the APA’s definition of mental health suggests that emotional well-being, which includes happiness, is an integral part of mental health (APA, 2023). Moreover, utilising a continuous scale is often used for evaluating happiness levels (see Blain & Rutledge, 2020; Rutledge et al., 2015; Rizzatto et al., 2022). As in these past studies, participants were not shown the numerical scale; instead, they used a slider labelled with ‘very unhappy’ on one end to ‘very happy’ on the other. For analysis purposes only, ‘very unhappy’ was assigned a value of -50, and ‘very happy’ a value of 50. It is important to note that our findings would be consistent even if these were reassigned other numbers to the scale. We now cite previous studies that have used such a scale in the manuscript to justify our selection (pg.15 &16).

- Why did you study the relationship Knowledge-Seeking and Mood only for negative information? Please, justify your choice. Without testing positive emotions your results might have a huge bias.

- Thank you for the opportunity to clarify. In both Studies 1 & 2, we did explore the association between participants' mood and positive and negative valence of webpages they selected to browse. Our findings indicated a significant and replicable link between mood and the negative valence of web content in both studies. In contrast, we did not observe a replicable relationship between mood and positive valence of web content (pg. 6-7). Having established the connection between mood and the negative valence of web content, we next aimed to investigate the causality of this relationship. Specifically, we wanted to understand whether a low mood leads individuals to seek out negative content and/or if exposure to negative content results in a negative mood. This prompted us to conduct Studies 3 and 4. Since we did not observe a relationship between positive valence scores and mood across Studies 1 & 2 (except in one test which did not replicate), there wasn't a case for examining the directionality of such a relationship (see pg. 6-7 & 9 for details). While it may have been interesting to also test for positive emotions despite the lack of replicable effects in Study 1 and 2, the choice does not introduce a confound or systematic bias that we can identify.

- I do not fully understand the rationale of your intervention. Why would providing people with cues about the valence of webpages alter their knowledge-seeking patterns? Do you mean with respect to the same topic?

- Yes, exactly – with respect to the same topic. A user enters a query and gets results on a Google search page. Those results can differ on a valence scale (some may be less negative than others for example). Normally a user is unaware of the valence of the content in the list of options. Providing labels informs the user about the content's valence and thus can alter their choice on which webpage to click on (pg. 11-12 & 18-19).

- Saying “three Google search result pages” is very broad. Are they controlled for topic? usually one might infer from the topic also the positive/negative content, how you did control for that?

- Precisely – they are controlled for topic. Each Google search results page featured *one* search query along with three genuine webpage links associated with *that same* query, from which participants could make a selection (pg. 11-12).

- I think that while you are stating this “Exposure to negative content was in turn associated with worse mood (controlling for pre-browsing mood). We established the causality of this relationship by exposing participants to either negative or neutral webpages.” You should also acknowledge the limitations of your work, like that you did not expose participants to positive content, and which the implications might be (e.g., it is likely that the experimental design introduced a bias in the results since you lack the comparison with positive emotions, and exposure with neutral is likely not going to increase mood)

- The comment made us aware that we were not sufficiently clear. Studies 1 and 2 show a relationship between negative content scores and mood: worse mood is related to more negative scores and vice versa. Studies 1 and 2 did not show a relationship between positive scores and mood (except one test that did not replicate). Thus, Study 3 examined the directionality and causality of *that* effect. It tests if negative content seeking *worsen* mood and/or if bad mood lead to negative content searches. The design does not introduce a bias (defined as systematic error or confound), because if positive content would have increased mood that would have only made the comparison to negative content easier to detect (as they would have effected mood in the opposite direction) as compared to neutral content (which does not affect mood). We now clarify this in the manuscript (pg. 9).

- Also, this is a strong statement “We found that participants' tendency to browse webpages of specific valence, either more negative or positive, was moderately stable over time, indicating that web-browsing behavior has both trait-like and state-like characteristics” and you just used ICCs to sustain this statement. So I would tone down that and again, explicitly consider limitations and what other data (e.g., longitudinal data, EMA) can offer.

- Thank you, we have now rephrased the sentence. The reason we conclude web-browsing behaviour likely has state-like and trait-like characteristics is because it was impacted by changes in mood (state) and also associated with mental-health questionnaires which are thought to be related (at least partially) to trait-like characteristics. We now state the limitation of our approach and also highlight what other data, such as longitudinal data, has to offer. In particular, we note that we only examine mood changes over five days. Extending the duration of data collection to weeks or months while employing ecological momentary assessment (EMA), will allow us to characterize the relationship between mental health and web-browsing patterns at a more granular level (e.g., by applying time-series analysis). (pg. 13).

- I think that the clear strength of the article is the analysis of the affective properties of text. I would encourage the authors to describe more clearly in both intro and discussion why such an approach is now needed, which are the theoretical and practical improvements, challenges, and how we can ameliorate such an approach.

- We appreciate the Reviewer's comment and agree that the methodological framework of our study is a notable strength. This framework combines psychological theory with computer science to explore the impact of web content's emotional properties on mental health. Such an endeavour is critical for enriching theoretical knowledge and aiding in the creation of practical interventions that either leverage or mitigate these effects. Our research not only examines these theoretical constructs but also introduces potential applications, such as monitoring mental health through web-browsing habits and online interventions. For instance, the intervention we investigated demonstrates that providing affective labels for web content in advance leads to less negative content being browsed which in turn significantly improves mood while browsing. We also highlight challenges, for example of evaluating the emotional content of webpages that primarily consist of images or videos rather than text. These topics are now articulated with greater clarity and detail in the revised manuscript, specifically in the Introduction (pg. 3) and the Discussion (pg. 13).

- Again, in the discussion you should refer your findings on “providing individuals with cues about the valence of webpages effectively changed their browsing patterns, leading to a decrease in exposure to negative content and an increase in exposure to positive content” to decision-making processes guided by emotions and related that more to existing theories that can explain your results.

- Thank you for this suggestion. There are existing theories on the pivotal role of emotions in information-seeking decisions, which seemed most relevant and we have now incorporated those into the introduction (e.g., Sharot & Sunstein, 2020; Kelly & Sharot et al., 2021; Lerner et al., 2015; Hockey et al., 2000; George et al., 2016; Paulus & Angela, 2012; Phelps et al., 2014; Pictet et al., 2011; Stigler, 1961; Karlsson et al., 2009; Charpentier et al., 2018) (pg. 3).

Methods

- Do you have other information on participants? SES, education level, race, country, etc

- Yes, we have income, education, ethnicity and English as one’s first language. These have now been incorporated into the analysis as covariates, in addition to age and gender. Importantly, all results remain consistent (see **Supplementary**). We also report these demographics for each Study (see **Supplementary**).

- The scales used to assess mental health (or mental ill-being) and the question on the happiness are unrelated theoretically to each other (happiness was not present in the baseline). Also, happiness refers to hedonic well-being, but only ill-being was comprehensively assessed. Be aware of the conceptual difference. Please, justify your choice.

- Thank you for the opportunity to consider our choice of terms. First, following the Reviewer’s earlier comment we now refer to ‘mental health’ rather than ‘well-being’. The exact definition of mental health differs across countries, scientists and organizations. The CDC states that “*Mental health includes our emotional, psychological, and social well-being*”; MIND – one of the largest mental health charities in the UK- states “*Mental health is a state of mental well-being*”. Almost all organizations, including the UK’s NHS, include in their definition “*mental health conditions*” or “*mental health problems*” listing psychopathology disorders. Thus, under the majority of definitions mental health problems are associated with psychopathology and mental health with a positive emotional state, which includes happiness and positive mood. Moreover, continuous low mood and unhappiness are symptom of depression and other affective disorders. Individuals with certain disorders (such as affective disorders) will experience low mood, which may be associated with searches of negative content. Further, frequent searches of negative content may lead to continuous low mood, which can trigger or enhance mental health problems. We now explain this in the manuscript (pg. 3).

- More control variables would be needed in the ANOVA.

- We have now included income, ethnicity, and education into the analysis as covariates, in addition to age and gender. Importantly, all results are unchanged (see **Supplementary**).

- In study 3 and 4, were participants randomized to the conditions?

- The approach we used in Studies 3 and 4, is to use the exact same advertisements for the different conditions, thus that participants could not tell them apart when deciding to participate (**pg. 17 & 18**). In other words, we did not randomly assign them to different conditions, but rather they unknowingly assigned themselves, with conditions being run approximately only 30min-120min apart (**see pg. 17 & 18**). As detailed in the response below, our checks confirm no difference between groups on any demographical characteristics (**see Supplementary**).

How did you check randomization?

- To check randomization, we assessed whether there were any differences between conditions with regard to demographic and socio-economic variables using a series of independent t-tests and chi-square tests. No significant differences were observed (**see Supplementary**).

Reviewer #2:

Remarks to the Author:

Dear Authors,

I was delighted to review the submitted manuscript “Knowledge-Seeking Reflects and Shapes Well-being”The manuscript reports on four related studies to investigate the (causal) link between the valence of browsed websites and affective well-being, and whether browsing activities can be manipulated by cues on the valence of website content. The authors found that exposure to negatively valenced online content is significantly linked to negative mood, and this relationship is bi-directional. Furthermore, providing labels on the valence of the website next to the links to the websites in a search engine promotes a more frequent selection of positively valenced and less negatively valenced website content.

The major strengths of this manuscript are 1) the use of different methodologies (observational and experimental), 2) the combination of self-reported well-being with browsed website content, on which automated sentiment analysis was performed, and 3) the multiple robustness checks conducted by the authors on the tools used for sentiment analysis, the samples of online content (whole website vs. random sections of a website; eye tracking), and the samples of study participants (participants with submitted browsing history during study period vs. those with submitted browsing history before study period).

- We thank the Reviewer for their positive assessment.

At the same time, I have several comments and questions that the authors should consider in a revised version of their manuscript:

1) Concerning the terminology used by the authors, I suggest substituting “knowledge-

seeking” with the more general term “information-seeking”. One can go even further and use the term “online browsing of web content” as the author did not limit the types of websites consumed by the study participants, and they may provide very different content from news to social support. Since only the valence, not the type of content, was considered in the studies, “online browsing of web content” is more coherent with what participants actually did online.

- This is an excellent suggestion. We now use the term information-seeking instead of knowledge-seeking, which we define as the active pursuit of knowledge (Sharot and Sunstein, 2020) and changed the title to use the term ‘web-browsing’.

2) Concerning the ecological validity of the findings, the authors should describe and discuss if they only considered web content browsed on a desktop or also mobile. Since people primarily go online via mobile devices these days, it is key to know if browsing behavior changes due to the mobile version of websites (e.g., How does attention change? How does the scrolling on a website change when done on the mobile screen compared to a desktop device?)

- In our initial studies participants could use any device they chose to and we did not record the type of device they used for browsing webpages. Thus, to address the Reviewer’s question we conducted two new checks to test whether the key parameters are different based on the device used. First, we asked one group of participants to view webpages on their smartphones and another group to view the same webpages on their desktop/ laptop computers. All were to rate the sentiment (positive and negative) of the pages. The results showed a remarkably strong positive correlation between the two groups’ ratings (positive scores: $r(23) = 0.980$, $p < 0.001$; negative scores: $r(23) = 0.995$, $p < 0.001$). This implies that the emotional impact of the webpages is comparable, regardless of the browsing device (see **Supplementary**).
- Second, we asked a new group of participants to browse the internet for 15-minutes on their cell phone on one day and on their desktop or laptop another day. We then calculated the valence scores of the webpages they selected to browse on each day. There was no significant difference in the valence scores between days (Negative score: Mean difference = -0.002, SD = 0.012, $t(27) = -0.672$, $p = .507$; Positive score: Mean difference = 0.001, SD = 0.035, $t(27) = 0.158$, $p = .875$). A Bayesian paired-samples t-test corroborated these findings, yielding a Bayes factor of 0.24 for negative and 0.21 for positive valence scores, indicating moderate evidence for the null hypothesis (see **Supplementary Materials**). Moreover, there was a significant correlation between the valence scores of information browsed on cell phones and lap/desktops (Negative score : $r(26) = 0.641$, $p < 0.001$; Positive score: $r(26) = 0.758$, $p < 0.001$).
- It is of course likely that attention patterns are different across devices, for example, time spent per page etc. The critical question for our specific study though is – do these lead to different emotional responses and/or to changes to the valence of webpages browsed? The new sets of data suggest the answer to both is no; neither the valence of the webpages that are freely browsed, nor their emotional impact, alters across devices.

3) If I understood it correctly, the authors used the browsing history of participants (i.e., the list of accessed websites) to access them at a later stage and extract all the text data for sentiment analysis. However, website content may change quickly, especially when

considering news websites. Though the authors conducted a robustness check by comparing participants with browsed websites before the actual data collection (including self-reports) and browsed websites during the actual data collection, and found no significant differences, I still wonder if exposure to primarily negatively or positively valenced web content is indeed stable.

- Being mindful that content can change quickly, we made sure to extract the vast majority of the text from webpages within 24-hours on average from the time the participant visited the page. However, it is still possible that some content may have changed during this time. Therefore we ran a new validation study to check for the stability of webpages' valence scores. To that end, we assessed the valence of 1000 random webpages on two consecutive days. Our findings revealed a very strong correlation between the valence scores on day 1 and 2 (negative scores: $r(998) = 0.912$, $p < 0.001$; positive scores: $r(998) = 0.902$, $p < 0.001$). This strongly suggests that content changing quickly is unlikely a concern effecting our results. We have now added this new result to the **Supplementary Materials**.

and to what extent it is determined by the algorithm of search engines. In other words, is website browsing user-driven (see selective exposure theories) or more likely provider-driven, or a combination of both?

- Algorithms likely impact the webpages participants expose themselves too. In particular, algorithms, which have been trained on participants past behaviour may work to perpetuate a participant's affective state by promoting specific types of information (such as negatively valenced content), thus exasperating the feedback loop discussed in the manuscript. Algorithms may also be a source of noise, in the sense that they alter peoples' natural search intentions. If that is the case, the relationship between mental health and self-driven searches is likely even greater than we report here (see **new Discussion, pg. 13**).

In this regard, I was also wondering if the website order in Study 4 was randomized to rule out order effects in the label condition.

- Yes, the order of the labels was randomised to rule out order effects. We now make this clear in the manuscript (**pg. 11 & 18**).

4) This brings me to another point concerning the perceived website valence. While preliminary checks identified a strong correlation between different procedures to evaluate the valence of web content for negatively valenced websites, the ICC for positively valenced websites was only moderate (see page 4. ICC = 0.499). Furthermore, the study findings were rather consistent and robust for negatively valenced web content and negative mood, but non-significant or only partly significant for positively valenced web content and positive mood. The differential effects have not been discussed sufficiently by the authors. Do they think this is a methodological artifact? Or an indicator of the complexity of positive mood/perceptions of positive content? Or maybe exposure time to positively valenced web content was not sufficient to detect a positive bi-directional effect on mood (see assumptions of a dose-response relationship)?

- Thank you for these insights. Indeed, while we observed a robust and replicable relationship between negative valence of pages browsed and mental health, the

insignificant finding for positive valence may either indicate a ground truth, or may be due to methodological difficulties the Reviewer lays out. We have now incorporated a new discussion of this in the Discussion section of our paper (pg. 13).

5) In addition to paying more attention to the findings on positive mood and positively valenced websites, it would also be interesting to learn more about the associations between well-being assessed on the three dimensions “Anxious-Depression”, “Social-Withdrawal”, and “Compulsive-Behaviour and Intrusive Thought” and website browsing. The authors only report on the overall well-being score across the three dimensions, but it would be interesting to learn if the relationships are larger for specific sub-dimensions, e.g., “Anxious-Depression”, especially considering that – as reported in the supplement – those with poor mental health (across the three dimensions) were more exposed to fear-related websites.

- We thank the Reviewer for this comment. A mixed ANOVA revealed no interaction between valence of web-browsing patterns and psychopathology dimensions (all P s > 0.27). In other words, the relationship is not significantly stronger for one dimension over the others. We now report this in the manuscript (pg. 6).

In addition to these 5 comments, I also have some minor suggestions regarding the structure/writing of the manuscript itself:

1) I suggest integrating the last paragraph of the results section in the discussion to avoid repetition.

- We have now integrated the last paragraph of the Results section in the Discussion (pg. 12).

2) In the discussion on p. 11, last paragraph, the authors should add some references to their statement “Previous research in this area has primarily focused on ...”

- We have now added the following references to support this statement:

Ayers, J. W., Poliak, A., Johnson, D. C., Leas, E. C., Dredze, M., Caputi, T., & Nobles, A. L. (2021). Suicide-Related Internet Searches During the Early Stages of the COVID-19 Pandemic in the US. *JAMA Network Open*, 4(1).

Gunnell, D., Derges, J., Chang, S. Sen, & Biddle, L. (2015). Searching for suicide methods. *Crisis*, 36(5), 325–331.

3) There are some minor grammar mistakes and typos that should be checked and resolved, e.g., in the abstract properties influence/alter instead of influences/alters; p. 5 NRC VAD lecion should be NRC VAD lexicon; p. 19 first sentence no comma before “knowledge?”

- Thank you for pointing our attention to these typos, we have now amended them.

4) Formatting of the title on p. 7, no “_”

- Thank you, we have now amended this typo.

Reviewer #3:

Remarks to the Author:

Summary:

This paper aims to examine how online knowledge-seeking affects well-being. Through four studies involving 947 participants, the researchers explore the connection between the affective properties of online information and individuals' psychological and emotional well-being. The results indicate that engaging with negative information online is associated with worse mental health and mood. By manipulating variables in the experimental setting, the researchers establish a causal reciprocal relationship between affective characteristics and well-being. Moreover, they devise an intervention that effectively modifies web-browsing habits to be less negative. These findings suggest that the type of knowledge individuals seek online reflects and influences their mental state, establishing a feedback loop that may contribute to ongoing mental health challenges. The study also proposes a potential approach for evaluating and improving welfare in the digital era by acknowledging the emotional impact of web content.

The paper is well-written and clearly structured. However, I have major concerns regarding the contribution and methodology of this work. Hence, I cannot recommend acceptance of this paper. Below I have listed some comments that may help the authors for future submissions of the paper.

- Thank you for your feedback. Below we address all the comments, which we believe has strengthened the paper.

Major Comments:

1. Novelty and positioning

A major concern with this paper is the very shallow discussion of related work. Neither in the Introduction nor in the Discussion section do the authors engage with the literature on mental health and online activity. As such, it is impossible to evaluate the contribution of this work over others that study online information-seeking (e.g., related to Covid, social networking sites, etc.) and mental health outcomes [1, 2, 3, 4, 5, 6]. Given that the paper does not disclose how it is situated in the existing literature, I have reservations as to what contribution the paper provides and whether it significantly advances our understanding of mental health and online activities.

- Following the Reviewer's comment, we have now expanded our review of the existing literature regarding online activity and mental health (**pg. 3**). The vast majority of existing research investigating the relationship between online activity and mental health has focussed on screentime (e.g., Babic et al., 2017; Page et al., 2010; Granic et al., 2014; Odgers, 2018). Attention has also been given to social media use, with studies looking at frequency of use (Orben & Przybylski, 2019; Brusilovskiy et al., 2016; Yoon et al., 2019; Seabrook et al., 2016) as well as scrolling vs active participation on social

media (Nisar et al., 2019; Escobar-Viera et al., 2018; Thorisdottir et al., 2019) and the type of information users share on social media (Guntuku et al., 2020; Valdez et al., 2020; De Choudhury et al., 2013; Kelley & Gillan, 2022; Eichstaedt et al., 2018). There are also studies relating prevalence of specific words in Google queries (such as ‘suicide,’ ‘therapist’) with indicators of the average well-being of a population (Ayers et al., 2021; Gunnell et al., 2015; Sueki, 2011; Hoerger et al., 2020; Ayers et al., 2012; Knipe et al., 2020; Misiak et al., 2020; Rana, 2020; Tran et al., 2017; Arora et al., 2019).

- Our work is empirically and theoretically distinct from all the above. We do not examine frequency of use, nor do we examine the content the user produces. Rather, we examine the *features* of the content they chose to *consume*. In particular, we focus on the *affective* properties of the content consumed. Our hypothesis is that because poor mental health is related to negative mood and thoughts, users will seek content related to those negative thoughts and feelings. In turn, this content will increase negative thoughts and feelings, forming a feedback loop. This is the first report of a (bidirectional and causal) association between the valence of content people select to browse online and their mental health. (To our knowledge this is also the first use of this technique (NLP of browsing history) and of an intervention that focuses on altering choices of the valence of content consumed. The new approach is critical for understanding how the content people browse impact their mental health and what can be done to facilitate or mitigate these effects. Thus, the new methods introduced and the findings, advance the theoretical landscape and contribute to the development of novel practical tools.

2. Methodological questions

The authors state that they recruit participants from Prolific, but where the specific inclusion criteria? Did participants have to speak English? Was the recruitment restricted to a specific geographic entity (e.g., US) or worldwide? In general, the authors should add more information on how participants were selected. For example, the authors should clearly state their inclusion or exclusion criteria. This information is vital for assessing the generalizability of the findings and ensuring the representativeness of the sample.

- We now detail the exclusion and inclusion criteria and demographic details in the Method section (pg. 14-18). Because our study was conducted in English participants in Studies 2-4 were recruited from the UK and US. In Study 1 due to a technical error 59 participants out of 312 were from other countries. We do not have a specific numerical breakdown for each specific country. Participants had to be 18 years or older. We did not have any other exclusion criteria. We also asked participants whether English was their first language and now added this covariate in the analysis with the results remaining unchanged (see **Supplementary**).

How was the sample size determined? I did not find a preregistration of that (which I would have loved to see). Compared to other studies testing e.g., behavioral interventions, I personally find the sample size somewhat small.

- For all studies we conducted pilot studies to determine sample sizes needed to observe effects of interest. Power analysis was run using the G*Power 3.1 software, with an

alpha of 0.05 and a 1-beta of 0.95. We have now added this information to the Methods section (pg. 14-18).

The paper states that users had to browse the web via Firefox. How was this asserted? Does this bias the results (e.g., to tech-savvy population)? I am surprised that this is not discussed as part of the limitations.

- The reason we asked subjects to use Firefox is that it was the only browser we were aware of that provides the details needed in the search history for us to be able to exert the essential data. We now explicitly state this in the manuscript (pg. 14). We did not assert that participants used Firefox, but if they did not then we would not have had the data needed to retrieve the content they browsed.
- The Reviewer asks whether the results might be specific to Firefox users. There is some evidence to suggest this is likely not the case. Specifically, in Study 4b, Firefox was not required and yet we observed that participants who engaged with more positive web content showed a significant mood improvement ($\beta = 0.090 \pm 0.023$ (SE), $t(160.74) = 3.941$, $p < 0.001$). This observation suggests that our findings might not be heavily influenced by the type of browser used. However, we now state in the discussion that Study 1-3 required participants to use Firefox and that future research would be beneficial in testing these findings across different web browsers (pg. 13).

How were social media pages handled? Note that the authors may get the link to the social media page as part of the browsing history but, due to security mechanisms, cannot crawl the actual content that is displayed. Hence, the “private” link will resolve to the landing page (e.g., facebook.com). I think this misses a major part of the actual online behavior. More crucially, this limitation is neither properly disclosed nor discussed (and not tested for as a robustness check). On the contrary, it may even lead to bias. As a thought experiment, let the landing page of facebook.com only contain negative words, then this may confound the analysis, so that the authors find that users with more social media consumption also experience lower well-being, even though this had little to nothing to do with the negative words (but maybe other parts on the social media page such as peer pressure etc.).

- Due to privacy concerns, participants were asked not to browse password-protected webpages, including social media pages. However, following the Reviewers’ comment we went back to the data and observed that a number of subjects did browse password-protected social media pages (Study 1: $N = 69$; Study 2: $N = 22$). To make sure that our results were not influenced by social media landing pages or web-browsing patterns that we could not assess, we excluded participants who engaged with password-protected social media platforms like Twitter, Facebook, and Instagram. Our findings (detailed in the Supplementary) remained unchanged after this adjustment. It would be interesting to generalize the results to include password protects webpages including social media in the future (see Discussion, pg. 13).

Figure 2 refers to data from $N=10$ participants. However, the point plot has way more than 10 points. Somethings is odd here (either the N is not correct, or the authors have multiple observations per person, which would imply that a different analysis is needed).

- Indeed, our analysis was for $N = 10$, with the plot showing all observations - that is each dot reflected one website, not one person. However, we understand the reviewer’s

point, which we understand as saying that a subject-based analysis is preferred. We thus increase the total number of participants to 19 and used a mixed linear model with fixed and random effects. Once again we find that the Positive Valence ($\beta = 0.304 \pm 0.051$ (SE), $t(19) = 5.986$, $p < 0.001$) and Negative Valence ($\beta = 0.342 \pm 0.163$ (SE), $t(19) = 2.105$, $p = 0.035$) of the text that participants attended to was significantly associated with that of the whole webpage. As before, this indicates that assessing the Valence of the whole text on a webpage is a good indicator of the valance of the partial text which the participant consumes. The new plot now shows one data point per participant (see **Figure 2 b&c**).

Likewise, it is unclear what confidence intervals are shown. 95%?

- Yes, they are indeed 95% confidence intervals. We have now made this explicit in the paper (**pg. 6**) and thank you for pointing this out.

The paper mentions that participants were instructed to browse the internet for studies 1 and 2 but are calm about further details (e.g. was there a suggested starting point). It is essential to clarify the specific guidelines given to participants to ensure consistency across the experimental conditions. In addition, I was wondering how the participants were introduced and debriefed to the experiment.

- We thank the Reviewer for prompting us to provide these additional details. We have now included the specific instructions in the **Supplementary Materials** and signpost the instructions in the Methods section (**pg.14**). Specifically, we did not provide a starting point for participants as we wanted the web-browsing to be as natural as possible. After completing the study, participants were thanked for their participation – there was no additional debriefing.

I would like to see more summary stats. As of now, the paper is mostly focused on the significance testing but some descriptives on the survey part, the sentiment variables, etc. would be helpful (e.g., I would like to see the mood variable distribution).

- Following the Reviewer's suggestion, we have now added additional descriptives to the **Supplementary Materials**.
-

It would also be interesting to see some examples of websites visited by the participants during the experiment (e.g., a table of the top 10 URLs).

- We agree that this would be interesting. However, due to our ethics protocol (what is known in the US as IRB), we are unable to report any specific URLs visited by participants, because of the risk of re-identification etc.

Did the authors allow individuals to participate in multiple studies or did they limit participation to just one study?

- Each subject participated in only one study. We now mention this in the Methods section (**pg. 14**).

Which software was used for the analysis?

- For statistical analysis we used a combination of R Studio and SPSS. For text pre-processing and quantification we used a combination of R Studio and Python. We now report this in the Methods section (pg. 16).

A minor point: did the participants gave informed consent. While this is not a big issue, I would recommend a statement, even if it is just a sub-clause.

- Yes, the participants gave informed consent. We have now included a statement about consent throughout the manuscript (pg. 14-18).

3. Generalizability in the field

The paper provides a host of evidence from the lab. However, the authors do not discuss how their results transfer to the field. Actively asking participants to browse may change their state of mind (e.g., make users adopt a utilitarian mindset instead of a hedonic mindset) and thus affect how they perceive web pages and what they search for. I would like to see additional explanations as to why the measured effect is transferable to the field. As an idea, instead of only checking the valence of the websites, the authors could, for example, check the frequency of URLs.

- To examine if actively asking participants to browse altered browsing behaviour, we conducted an additional analysis. Specifically, while subjects were asked to browse the web during the experiment and send us their browsing history, some participants did not follow these instructions and instead submitted archived browsing history. This means that participants provided their 'natural' browsing history from a period prior to the study session - a time during which they browsed as they usually would. This deviation was identified through notably short study completion times and was then explicitly confirmed by the subjects themselves (i.e., we explicitly asked participants at the end of the study). We then tested if there were any differences in the valence scores of the webpages browsed between these two groups ('in-study' data vs. 'natural' achieved data) – there was none, neither for Positive scores (in-study data: $M = 0.152$, $SD = 0.032$; archived data: $M = 0.154$, $SD = 0.030$; $t(608) = 0.416$, $p = 0.678$, Cohen's $d = 0.030$) nor for Negative scores (in-study data: $M = 0.031$, $SD = 0.013$; archived data: $M = 0.032$, $SD = 0.016$ $t(608) = 0.691$, $p = 0.498$, Cohen's $d = 0.012$). This analysis suggests that the study's setup did not bias participants towards browsing webpages with a specific emotional valence. Therefore, the findings are likely transferable to natural browsing behaviour. We now report this analysis in the Results section (see pg. 7-8).

Relatedly, the authors let participants “browse” two web pages in the first part of study 3. Does this really reflect a typical browsing activity where I would assume that people randomly visit multiple websites? Could it be that the effect measured for mood is solely due to the negative content rather than the activity itself?

- Yes, precisely. The goal of the first part of Study 3 was to test if negative content leads to negative mood. This was done in order to tease apart the directionality of the correlation observed in Studies 1 and 2 between negative mood and browsing negative pages to its basic components. The goal was not to reconstruct a typical browsing activity (which is already tested in Studies 1 and 2), but rather to test in a very controlled setting whether simply exposing a subject to (just two) negative websites will affect

mood. We found the answer is yes. That result then helps interpret the correlation observed in Studies 1 and 2. In particular, it suggests that exposure to negative content leads to negative mood. We also found in the second part of Study 3 that negative mood subsequently leads to negative browsing.

The authors are also calm about the devices used during the experiment. I guess that many people would nowadays browse with mobile devices. Did all participants use the same type of device when conducting the experiment? If not, have the authors checked how effects vary between mobile and stationary devices? This could give several interesting analyses for the supplements.

- In our initial studies subjects could use any device they wished and we did not record the type of device used by participants for browsing webpages. Thus, we conducted two new studies to test whether the key parameters of the study are different based on the device used. First, we asked one group of participants to view webpages on their smartphones and another group to view the same webpages on their desktop/ laptop computers. All rated the sentiment (positive and negative) of the pages. The results showed a remarkably strong correlation between the two groups' ratings (Positive scores: $r(23)=0.980$, $p < 0.001$; Negative scores: $r(23) = 0.995$, $p < 0.001$). This implies that the emotional impact of the webpages is comparable, regardless of the browsing device (see **Supplementary**).
- Second, we asked a new group of participants to browse the internet for 15-minutes on their cell phone on one day and on their desktop or laptop another day. We then calculated the valence scores of the webpages they selected to browse on each day. There was no significant difference in the valence scores between days (Negative score: Mean difference = -0.002 , $SD = 0.012$, $t(27) = -0.672$, $p = .507$; Positive score: Mean difference = 0.001 , $SD = 0.035$, $t(27) = 0.158$, $p = .875$). A Bayesian paired-samples t-test corroborated these findings, yielding a Bayes factor of 0.24 for negative and 0.21 for positive valence scores, indicating moderate evidence for the null hypothesis (see **Supplementary**). Moreover, we observed a significant correlation between the valence scores of information browsed on cell phones and lap/desktops (Negative score: $r(26)=0.641$, $p < 0.001$; Positive score: $r(26)=0.758$, $p < 0.001$).
- Together, these new sets of data suggest that neither the valence of the webpages that are freely browsed, nor their emotional impact, alters across devices.

4. Questionable labeling in Study 4

I am wondering why the authors chose to label the intervention “feel better”, “feel worse” and “neutral”. These labels seem to indicate an action associated with the choice of a certain webpage rather than an assessment of the web page’s valence. I am concerned that this might have introduced a bias in the choice by participants since it seems unlikely somebody is going to click on a webpage labeled as “feel worse.”

- Following the Reviewers' question we now clarify the reasoning behind the labelling choice. The labelling was an intentional design choice aligned with the core aim of our study, which was to investigate whether we could *induce* a selection bias. Our hypothesis was that making participants explicitly aware of the *potential emotional*

impact of the information on webpages would influence their web-browsing patterns. The labels thus served as a direct manipulation to test if participants would exhibit a preference or avoidance behaviour when presented with such cues. The idea that participants might be unlikely to click on a webpage labelled as '*feel worse*' is precisely the type of behaviour we intended to measure. Note, however, that still participants did select those webpages on 24.3% of the trials. We now clarify this objective in the revised manuscript (pg. 11).

A crucial extension of study 4 would be to examine the impact of the intervention on participants' well-being. While the study demonstrates the intervention's positive influence on browsing behavior, it remains unclear how this intervention affects the resulting change in well-being. Exploring this aspect would provide a comprehensive understanding of the intervention's effectiveness. As a case in point, such assessment is crucial to understand the effect size of the intervention and to judge whether it is meaningful in practice.

- This is an excellent suggestion. Following the Reviewer's suggestion we conducted a follow up study (i.e., Study 4b). The study was similar to Study 4a (previously termed Study 4), except that we asked participants to indicate their mood before and after engaging with the intervention. Once again, when engaging with the intervention participants selected to browse more positive webpages ($M\% = 47.75$, $SD = 25.17$) than negative ($M\% = 24.35$, $SD = 20.02$, $t(199) = 8.041$, $p < 0.001$) or neutral ($M\% = 27.90$, $SD = 19.35$, $t(199) = 6.986$, $p < 0.001$). Importantly, when engaging with more positive webpages participants' mood improved ($\beta = 0.090 \pm 0.023$ (SE), $t(160.74) = 3.941$, $p < 0.001$). We now report this study in the main text (pg. 11-12).

5. Measures and methodological approach

For the calculation of Psychopathology Dimension Scores (in studies 1 and 2), clarification is needed on what these terms entail and how they are measured. In addition, it is unclear which questionnaire results (pre- or post-web-browsing) were used to calculate the three psychopathology dimension scores. This detail is crucial for understanding the associations between web browsing and psychopathology.

- We thank the Reviewer for pointing out that the measuring of the three psychopathology dimensions was not clear enough. The calculation of psychopathology dimension scores in Studies 1 and 2 follow the methodology outlined in Seow & Gillan (2020) and Kelly & Sharot (2021) exactly. First, each questionnaire item rating is Z-scored across participants. Second, each Z-score for each item is multiplied by its factor weights. These weighted scores are then summed for each participant and dimension. Factor weights were originally identified by Gillan et al. (2016) by conducting a factor analysis across psychopathology questionnaires. They found that three factors best represented the underlying structure of psychopathology, capturing the breadth and complexity of symptoms across these disorders. This dimensionality approach is beneficial over traditional psychopathology classifications as it allows for the identification of symptoms that may be indicative of multiple psychiatric conditions, thereby providing a more nuanced understanding that crosses conventional clinical categories (Cuthbert & Insel, 2010; Cuthbert & Insel, 2013). This has now become a well-established approach in the field (Gillan et al., 2016; Rouault et al., 2018; Seow & Gillan, 2020; Kelly & Sharot, 2021; Cuthbert & Insel, 2010;

Cuthbert & Insel, 2013). We now detail the calculation method in the Methods section (pg. 16).

- To generalize our results across different methodological protocols, in Study 1 psychopathology assessments were carried out before the participants engaged in web browsing and in Study 2 after. This ensures that the effects are not contingent on a specific order of measures taken. We now make this clear in the manuscript (pg. 4-9 & 17).

I am further wondering why the authors chose to include the psychopathology features in their analysis. At the moment, they are not included consistently in the study. The authors measure them in studies 1 and 2 but chose not to intervene on them in studies 3 and 4. Why are the psychopathology features not intervened in studies 3 and 4?

- Thank you for prompting us to clarify. Both psychopathology symptoms and mood are associated with mental health and mental health conditions, according to the CDC, NHS and others. One difference is that while psychopathology scores are thought to be relatively stable (Prenoveau et al., 2011; Knowles et al., 2020), mood can be transient. At the same time, however, constant low mood is a symptom of depression and other affective disorders. While it is relatively easy to manipulate mood in a period of minutes, it is difficult to do so for psychopathology. As the purpose of Studies 3 and 4 was to test for causation via manipulation, it made the most sense to manipulate mood. The notion is that changes in mood could eventually impact psychopathology symptoms (of which mood in itself is one). We now clarify this in the revised manuscript (pg. 9).

The authors chose self-reported “Mood” as a second dependent variable measured on a scale from -50 to + 50. The authors should provide additional information on why they chose this scale and how reliable it is. Relatedly, how did the authors set the endpoints and how does this influence the results? Could this scale be too fine-grained for participants (e.g., what is the difference between 30 and 31 but also between 30 and 40)?

- The comment made us realize that the scale was not clearly described. The scale was a slider labelled with '*very unhappy*' on one end and '*very happy*' on the other – *no numbers were shown at all*. This type of scale is commonly used to measure happiness and mood (e.g., Blain & Rutledge, 2020; Rutledge et al., 2015; Rutledge et al., 2015; Rizzatto et al., 2022). For analysis purposes only, '*very unhappy*' was assigned a value of -50, and '*very happy*' a value of 50. These numbers are arbitrary and findings are consistent if other numbers would be assigned. We now add this information to the Method section (pg. 8, 15-17).

I am struggling with the consistency of the methods. In some places, the authors employ linear regression analysis but ANOVA in others. This makes it hard to compare the results. To streamline the methodological approach for the studies, I strongly advise employing a regression analysis approach rather than comparing two groups using ANOVA in study 3.

- Following the Reviewers' recommendation, we conducted a linear regression in place of the ANOVA in Study 3. Specifically, we ran a linear regression to predict post browsing mood from condition (0 = control condition, 1 = negative valence condition), controlling for pre-browsing mood, age, and gender. Condition was found to be a

significant predictor of post browsing mood ($\beta = -0.503 \pm 0.066$ (SE), $t(101) = -5.034$, $p < 0.001$, partial eta squared = 0.151), indicating that subjects in the negative valence condition experienced lower post browsing mood (estimated marginal mean = 6.505, SE = 3.059) relative to subjects in the neutral condition (estimated marginal mean = 4.787, SE = 2.787; see **Figure 5a**). We have now added this to the Results section (**pg. 9-10**).

In the regression equation, the valence score (ranging from negative to positive) of web pages could be incorporated as a single continuous variable. This would enable the interpretation of the coefficient of this variable in a causal manner.

- Thank you for this suggestion. As one of our aims was to characterize the impact of positive and negative content separately on mental health, we opted to maintain the separation of these variables in the analysis, as to provide the most complete characterization of the data and results.

Moreover, considering the change in well-being (post vs. pre-browsing) as the dependent variable would provide more valuable insights and easier interpretation.

- Thank you for this suggestion. Below the Reviewer also suggests predicting post browsing mood while controlling for pre-browsing mood (see two comments below), we adopted this latter approach throughout the article for consistency.

Similarly, for study 4, I would recommended utilizing a regression analysis approach, treating the variable of interest as the treatment assignment to the label condition versus the no-label condition. (As a result of the regression analyses, they could also run an additional check where they control for browsing time, as the difference in browsing time may explain also differences in the dependent variable.)

- Following the Reviewers' recommendation, we conducted separate regressions for Study 4 (now Study 4a), imputing the condition (0 = no label condition, 1 = label condition) as the independent variable and each of the three labels (i.e., negative, neutral and positive) as separate dependent variables. Participants in the label condition showed a tendency to choose webpages with positive labels more frequently ($\beta = 0.390 \pm 0.168$ (SE), $t(109) = 2.322$, $p = 0.022$) and avoided those with negative labels ($\beta = -0.370 \pm 0.151$ (SE), $t(109) = -2.451$, $p = 0.016$) compared to those participants in the no label condition. There was no effect of condition on selecting neutral webpages ($\beta = -0.002 \pm 0.010$ (SE), $t(109) = -0.992$, $p = 0.992$, see **Figure 6b**). We have now added these regression analyses to the Results section (**pg. 11-12**).
- Regarding browsing time, we do not have timestamps for participants browsing of webpages. We state this in the Discussion section (**pg. 13**).

Also, I suggest the authors streamline their usage of dependent variables to one per category. For example for the mood as the dependent variable, I would highly suggest the authors either always use (a) the delta in the well-being (i.e., post-browsing mood - pre-browsing mood) or (b) the post-browsing mood and a control for the pre-browsing mood. This would greatly help the comparability of the different studies.

- Thank you for this suggestion. We now follow the Reviewer’s recommendation (b) and predict post browsing mood while controlling for pre-browsing mood in all relevant analyses.

An important part of well-being is positivity. The authors have analyzed this in studies 1 and 2 but not in study 3. I was wondering why. It seems like it would have been natural to check how browsing positive websites affects mood and whether a positive mood leads to positive browsing as indicated in studies 1 and 2. Why have the authors omitted this in study 3? (I think that this, in general, could be a nice addition for a supplement.)

- In light of the Reviewer’s comment, it became apparent that we did not clearly explain why we conducted Study 3 and how Study 3 was informed by Studies 1 and 2. In Studies 1 and 2 we did not observe a replicable relationship between mood and positive scores for web browsing. Nor did we observe a replicable relationship between mental health and positive scores for web browsing. In contrast, we did find a replicable relationship between negative scores of web browsing valence and both mood and mental health. Thus, the next stage was to test whether the latter relationship (negative scores and mood) was causal. As no consistent significant relationship was found between positive scores and mood there was no motivation to test for causality of a non-replicable effect. We now clarify this in the revised manuscript (pg. 9).

6. Clearer discussion of limitations and findings

Moreover, a more detailed discussion of the limitations of the study design is needed. For example, participants might have chosen to browse only certain web pages or cleaned their search history post-hoc since they knew it would be viewed. In addition, self-reported emotions might be biased as well.

- We thank the Reviewer for their comment. Indeed, if some participants cleaned their search history or did not answer questionnaires accurately, our data would include noise not accounted for, and thus the true effects would be even larger than reported in the manuscript. We now discuss this and other limitations in the Discussion (pg. 13). Moreover, we also mention that future improvements can include adding analysis of images and videos; collecting timestamps of participants’ web-browsing to measure the exact amount of time users spend on each piece of content; including password-protected websites such as social media platforms; including browsers beyond Firefox, and extending the duration of data collection to weeks or months while employing ecological momentary assessment (EMA; pg. 13).

How do the findings add to existing research? Others have found that negative emotions on social media are contagious?

- We now expanded our review of the existing literature regarding online activity and mental health (pg. 3). The vast majority of existing research investigating the relationship between online activity and mental health has focussed on screentime (e.g., Babic et al., 2017; Page et al., 2010; Granic et al., 2014; Odgers, 2018). Attention has also been given to social media use, with studies looking at frequency of use (Orben & Przybylski, 2019; Brusilovskiy et al., 2016; Yoon et al., 2019; Seabrook et al., 2016) as well as scrolling vs active participation on social media (Nisar et al., 2019; Escobar-Viera et al., 2018; Thorisdottir et al., 2019) and the type of information users share on

social media (Guntuku et al.,2020; Valdez et al., 2020; De Choudhury et al., 2013; Kelley & Gillan, 2022; Eichstaedt et al., 2018). Studies have also related prevalence of specific words in Google queries (such as ‘suicide,’ ‘therapist’) with indicators of the average well-being of a population (Ayers et al., 2021; Gunnell et al., 2015; Sueki, 2011; Hoerger et al., 2020; Ayers et al., 2012; Knipe et al., 2020; Misiak et al., 2020; Rana, 2020; Tran et al., 2017; Arora et al., 2019).

- Our work is empirically and theoretically distinct from the above. We do not examine frequency of use, nor do we examine the content the user produces. Rather, we examine the *features* of the content they *chose* to *consume*. In particular, we focus on the *affective* properties of the content consumed. Our hypothesis is that because poor mental health is related to negative mood and thoughts, users will select to consume content related to those negative thoughts and feeling. This content in turn will increase negative thoughts and feelings, forming a feedback loop. This is the first report of a (bidirectional and causal) association between the valence of content people select to browse online and their mental health. To our knowledge this is also the first use of this technique (NLP of browsing history) and an intervention that focuses on altering choices of valence of content consumed. The new approach is critical for understanding how the affective properties of content people browse reflect and shape their mental health and what can be done to facilitate or mitigate these effects. Thus, the new methods and findings introduced, advance the theoretical landscape and contribute to the development of novel tools.

Are the authors effect sizes of similar magnitude? In general, I would like to see more discuss of the effect size and whether they are meaningful? (as of now, the authors appear to primarily search for “significance” but with little focus on the actual impact).

- This is an excellent suggestion. We now consistently report effect sizes. The key findings are of effect sizes ranging from 0.12-0.43 across our studies and are now reported in the main text and **Supplementary**. These effect sizes are comparable to those reported in previous studies. For instance, Twenge and colleagues (2017) identified a significant correlation ($r = 0.05$) between social media screentime and depressive symptoms. Similar studies, including those by Yoon and colleagues (2019) with $r = 0.11$, McCrae and colleagues (2017) with $r = 0.13$, and Vahedi and Zannella (2019) with $r = 0.17$, have also reported significant correlations between social media use and mental health symptoms. Additionally, in the context of sharing negative emotional content online, Kelley and Gillan (2022) found a significant correlation of $r = 0.17$ between the negativity of content shared on Twitter and the severity of depression.

Validation of experimental setup

7. Lexicon-based Emotion Detection

The authors validate the use of the NRC lexicon for quantifying the affective properties of web pages by (a) comparing it to other dictionary-based as well as machine-learning methods and by (b) validating it using a comparison to human annotation, which is commendable. However, information about the interclass correlation coefficient (ICC) between the machine learning approach (Distilbert) and human assessment would provide more justification and validity in using the NRC lexicon, as human annotation

should be regarded as the gold standard. Also, it would be beneficial to elaborate on the potential limitations of the selected lexicon.

- The Reviewer's insightful recommendation led us to evaluate the Intraclass Correlation Coefficient (ICC) between the machine learning technique (Distilbert) and human assessment. The results showed a weaker correlation (Negative score: ICC = 0.472, $p < 0.001$; Positive Score: ICC = 0.384, $p < 0.001$) compared to the ICC between NRC and human assessment (Negative Score: ICC = 0.707, $p < 0.001$; Positive score: ICC = 0.499, $p < 0.001$). This indicates that in our study, the NRC method more accurately reflects human ratings than the machine learning approach. We have now incorporated this finding into the paper (pg. 5). In terms of limitations - the lexicon approach does not incorporate context consideration. Nevertheless, in this instance it outperformed the machine learning approach based on large language model (LLM), and thus was selected. We state this in the revised manuscript (pg.13).

Why was the ICC only computed for a subset of 100 participants?

- A power analysis on a smaller subset revealed $N=100$ would provide sufficient power. Thus, to save resources 100 were run. We now explain this in the manuscript (pg. 15).

How was the sentiment variable aggregated over different pages (e.g., sum, average)?

- The sentiment scores from webpages were averaged for each participant (pg. 5 & 14).

8. Minor Comments:

Reliability of Browsing History Submission: The paper states that participants were asked to confirm the accuracy of their browsing history submissions. As an alternative, one could consider other methods, such as checking timestamps within the browsing history, to verify the validity of the submissions.

- We thank the Reviewer for their comment. We do not have timestamps for participants browsing of webpages (see pg. 13).

The authors might want to consider restructuring parts of the paper. On the one hand, I would suggest removing the validation studies from the results section and instead adding a subsection “validation” to the methodology section. On the other hand, authors might want to add tables with the results from the regression and/or ANOVA analysis to provide a clearer overview.

- In response to the Reviewer's suggestions, we have created a dedicated table for the Supplementary ANOVA and regression analyses in the **Supplementary section**. This addition provides a clearer overview. In the main text, however, we continue to report these analyses in paragraph form to maintain consistency across different studies and types of analysis. Furthermore, we have now prominently included a subtitle for the validation section on pg. 5. Given the importance of this section for understanding and interpreting the study's findings, we chose to keep it in the results section to ensure it captures the reader's attention. However, the methodologies of all validation studies are now thoroughly discussed in the **Methods** section (pg. 14-15).

References

- [1] Charpentier, C. J., Cogliati Dezza, I., Vellani, V., Globig, L. K., Gädeke, M., & Sharot, T. (2022). Anxiety increases information-seeking in response to large changes. *Scientific Reports*, 12(1), 7385.
- [2] Braghieri, L., Levy, R. E., & Makarin, A. (2022). Social media and mental health. *American Economic Review*, 112(11), 3660-3693.
- [3] Wilding, S., O'Connor, D. B., Ferguson, E., Wetherall, K., Cleare, S., O'Carroll, R. E., ... & O'Connor, R. C. (2022). Information seeking, mental health and loneliness: Longitudinal analyses of adults in the UK COVID-19 mental health and wellbeing study. *Psychiatry research*, 317, 114876.
- [4] Toseeb, U., & Inkster, B. (2015). Online social networking sites and mental health research. *Frontiers in psychiatry*, 6, 36.
- [5] Bell, V. (2007). Online information, extreme communities and internet therapy: Is the internet good for our mental health?. *Journal of mental health*, 16(4), 445-457.
- [6] Frison, E., & Eggermont, S. (2017). Browsing, posting, and liking on Instagram: The reciprocal relationships between different types of Instagram use and adolescents' depressed mood. *Cyberpsychology, Behavior, and Social Networking*, 20(10), 603-609.

- Thank you. We have now added those references that we found the most relevant to the Introduction.

Requests	Response
R3 request: I recommend to include another robustness check on how omitting the 59 participants not from the US or UK affects the results. You indicated that: We have now carried out this additional robustness check and all results remain the same (see Supplementary Analysis 7). However, there is an inconsistent finding for Browsing negative text relates to worse mood and vice versa (SI, P6), where $p = .052$. We would request that: Not to describe the results as being the same, given the inconsistent finding. Please note this inconsistent finding and, if you wish, add information on the related effect sizes and confidence intervals. In addition, not to describe this result as ‘marginally significant’. Instead, please describe it as not statistically significant and note the effect size and confidence intervals if you wish to compare the findings to the main analyses.	As advised by the Editor, we have now removed this analysis.
Relatedly, “We found that participants who reported better mood prior to browsing the internet, exposed themselves to less negatively valenced webpages ($\beta = -0.128 \pm 0.066$ (SE), $t(222) = -1.952$, $p = 0.052$”.	As advised by the Editor, we have now removed this analysis.
P values that exceed the conventional level of statistical significance (i.e., 0.05) are simply not statistically significant. Report these tests as not statistically significant and refrain from making any claims in support of the hypotheses tested based on ‘marginally significant’ results. Therefore, a) Please revise L358 – 359: “They also tended to select webpages with positive (“feel better”) labels more than participants in the no label condition” (... $p = 0.052$). b) Please revise the interpretation accordingly in this section and following sections where relevant e.g., Discussion). c) Please also revise “results remain consistent after controlling also for ethnicity, income, education and primary language (see Supplementary Analysis 5)”,	(A-C) We have now amended our language in the manuscript.

as there is an significant effect when controlling for the demographics "... reported in Table 3, participants in the label condition browsed more positive labels ($\beta = 0.364 \pm 0.182$ (SE), $t(102) = 3.179$, $p = 0.048$)". Please revise for accuracy.	
Please separate the Data availability and Code availability sections, with the Code Availability section placed after the Data Availability section.	We have now separated these two sections.
Please list IRB protocol approval numbers for each study (or a single protocol number if approval was obtained for the project as a whole).	
Please revise "CI []" to "'95% CI []"	We have now made this addition.
Please add the statistics used (e.g., linear regression) and sample sizes for Figures 3 and 4 in the figure captions.	We have now added this content.
L277: was not statistically significant in Study 2	We have now added the word 'statistically' to the statement.
My apologies for not making it explicit in our previous request. Thank you for explaining in the Reporting Summary. Please ensure that the Methods section includes a statement on randomization. Indicate whether the data collection was randomized or appropriately blocked, how samples were assigned to the various experimental groups and whether there was any randomization in the organization of the experimental conditions or stimulus presentation.	Thank you for your comment. There are no conditions nor groups etc in Study 1, 2, and study 4b, thus there is no "assignments to various experimental groups". Neither is there any stimuli in Exp 1 and 2. Therefore, this comment is only relevant for Exp 3 and 4, where we now include relevant statements.
In addition, please ensure that the Methods section includes a statement indicating whether blinding was used, which applies to Studies 3 and 4.	Thank you for your comment. There is no groups nor conditions in Exp 4b. Moreover, the word "blinding" is irrelevant/inappropriate in the context of Exp 3 and 4a. In one condition participants received labels (or negative webpages to read) and in the other they did not receive labels (or neural webpages to read). The point of the study is to ask -how do labels (or valence webpages) impact behaviour? Subjects are obviously aware that they are receiving labels (or that the valence of webpages are negative), that is the point of the intervention. If the intention in

	this comment is to say – did they know there was another condition - then the answer is no they were not aware, which we now state in the paper.
In response to R2's comments regarding mental health and mood: We have now cited relevant literature and expanded on our discussion to support the claim that mood is a feature of mental health (pg. 7). However, there is no such discussion on pg.7. Please add the literature and discussion to support the claim.	Apologies for the inaccurate signposting. We discuss this in the introduction (p.g., 3) and also in the methods (p.g., 11).
Reporting Summary: In the main text, you indicated that “we used a combination of R Studio and Python (L636)”. Please revise the Reporting Summary accordingly in the Software and Code section.	We have made this change.